# Multivariate Stochastic Dominance via Optimal Transport and Applications to Models Benchmarking

**Gabriel Rioux**
Center for Applied Mathematics
Cornell University

**Apoorva Nitsure**
MIT-IBM Watson AI Lab
IBM Research

**Mattia Rigotti**
MIT-IBM Watson AI Lab
IBM Research

**Kristjan Greenewald**
MIT-IBM Watson AI Lab
IBM Research

**Youssef Mroueh**
MIT-IBM Watson AI Lab
IBM Research

## Abstract

Stochastic dominance is an important concept in probability theory, econometrics and social choice theory for robustly modeling agents' preferences between random outcomes. While many works have been dedicated to the univariate case, little has been done in the multivariate scenario, wherein an agent has to decide between different multivariate outcomes. By exploiting a characterization of multivariate first stochastic dominance in terms of couplings, we introduce a statistic that assesses multivariate almost stochastic dominance under the framework of Optimal Transport with a smooth cost. Further, we introduce an entropic regularization of this statistic, and establish a central limit theorem (CLT) and consistency of the bootstrap procedure for the empirical statistic. Armed with this CLT, we propose a hypothesis testing framework as well as an efficient implementation using the Sinkhorn algorithm. We showcase our method in comparing and benchmarking Large Language Models that are evaluated on multiple metrics. Our multivariate stochastic dominance test allows us to capture the dependencies between the metrics in order to make an informed and statistically significant decision on the relative performance of the models.

## 1 Introduction

In choice theory, economics, finance, and models benchmarking, agents are faced with stochastic prospects that are (eventually) multivariate random variables which they wish to order according to a utility or risk measure of interest. To formalize such a notion of ordering of stochastic quantities, the concept of stochastic dominance can be utilized.

In the univariate case, a standard notion of order is given by *First order Stochastic Dominance* (FSD) which can be expressed in terms of the quantiles of the underlying random variables [Ogryczak and Ruszczynski, 2002]. To wit, a random variable $X$ dominates another random variable $Y$ in FSD, if it has larger quantiles than $Y$ across all percentiles. A weaker notion of FSD, called almost stochastic dominance was introduced in del Barrio et al. [2018]. Their approach is based on optimal transport (OT) and consists of measuring a ratio which quantifies how close $X$ is to dominating $Y$. del Barrio et al. [2018] further lay the groundwork for principled statistical analysis of almost FSD by establishing a central limit theorem for the empirical ratio as well as consistency of the bootstrap. Dror et al. [2018], Ulmer et al. [2022], Nitsure et al. [2023] used the almost FSD testing framework in benchmarking Large Language models to make statistically significant decisions regarding which model to select when these models are evaluated with a metric of interest on test data.

38th Conference on Neural Information Processing Systems (NeurIPS 2024).

Our main motivation is to extend the testing framework for almost FSD of del Barrio et al. [2018] to the multivariate case as to enable applications with dependencies between metrics. For instance, the problem of multivariate portfolio selection in financial applications has been treated via a reduction to univariate orders [Kouaissah, 2021]. Another application of interest is that of multi-metric benchmarking which is central nowadays in ranking and selecting Large Languages Models [Bommasani et al., 2023, Chang et al., 2023, Huang et al., 2023, MosaicLM, 2023, Wolf, 2023, Zhang and Hardt, 2024]. Current approaches such as Nitsure et al. [2023] use aggregation techniques to reduce the ordering to the univariate case thereby ignoring dependencies between metrics.

Our starting point is the so-called *standard multivariate stochastic order* (see Chapter 6 in Shaked and Shanthikumar, 2007). This order can be expressed in terms of couplings between random vectors. This insight allows us to follow the approach of del Barrio et al. [2018] and define an almost multivariate FSD via OT in Section 2. This notion of stochastic dominance can be defined using multivariate violation ratios that are expressed as optimal transport problems with smooth costs [Manole and Niles-Weed, 2024, Hundrieser et al., 2022, Groppe and Hundrieser, 2023]. Given that empirical OT suffers from the curse of dimensionality, we resort in Section 3 to entropic regularization [Cuturi, 2013] to alleviate that issue and hence define Entropic Multivariate Violation Ratios. We establish in Section 3 convergence of these entropic violation ratios as the regularization parameter goes to zero, as well as a central limit theorem and bootstrapping consistency in Section 4 using the functional delta method [Römisch, 2006]. We highlight that the delta method has seen general success for proving limit theorems with entropic OT [Hundrieser et al., 2024, Goldfeld et al., 2024a,b]. Armed with this theory, we propose a new framework for hypothesis testing of multivariate stochastic dominance and apply it to multi-metric benchmarking of LLMs. Multivariate FSD captures the dependencies between the metrics in this setting and leads to a more robust ordering.

**Notation** The indicator function of a set $A \subset \mathbb{R}^d$ is denoted $\mathbb{1}_A(x)$ and takes the value 1 if $x \in A$ and 0 otherwise. We also adopt the following shorthand notation, $\mathbb{R}_+ = [0, \infty)$, the maximum of two numbers $a, b \in \mathbb{R}$ is denoted by $a \vee b$, and for vectors $x, y \in \mathbb{R}^d$, we write $x \leq y$ to indicate that $x_i \leq y_i$ for every $i \in \{1, \ldots, d\}$.

Throughout, $\mathcal{P}(\mathbb{R}^d)$ is the set of all probability measures on $\mathbb{R}^d$. A measure $\eta \in \mathcal{P}(\mathbb{R}^d)$ is said to be sub-Gaussian with parameter $\tau^2 \geq 0$ with respect to the 1-norm provided $\mathbb{E}_\eta \left[ \exp(\|X\|_1^2 / 2\tau^2) \right] \leq 2$.

Convergence in distribution of random variables is denoted by $\overset{d}{\to}$ (in the sense of Hoffmann-Jørgensen when necessary, see Chapter 1 in van der Vaart and Wellner, 1996).

## 2 Optimal transport and stochastic order

### 2.1 FSD and Almost Stochastic Dominance in One Dimension

To properly motivate our results on multivariate FSD, we first review some theory for the univariate setting. For random variables $X, Y$, it is said that $X$ dominates $Y$ in the stochastic order (denoted $X \underset{\text{FSD}}{\succcurlyeq} Y$) if $\mathbb{P}(X \leq t) \leq \mathbb{P}(Y \leq t)$ for every $t \in \mathbb{R}$. Formally, this means that the inequality $Y \leq X$ generally holds for a given instantiation of these random variables. This condition can be cast, equivalently, as $F_Y^{-1}(t) \leq F_X^{-1}(t)$ for every $t \in (0, 1)$, where $F_X^{-1}(t)$ and $F_Y^{-1}(t)$ are the quantile functions for $X$ and $Y$ respectively. With this formulation in mind, del Barrio et al. [2018] propose the following index of almost stochastic dominance;

$$\varepsilon_{\mathcal{W}_2}(\mu, \nu) = \frac{\int_0^1 (F_Y^{-1}(t) - F_X^{-1}(t))_+^2 \, dt}{\mathcal{W}_2^2(\mu, \nu)}, \text{ where } \mathcal{W}_2^2(\mu, \nu) = \int_0^1 (F_X^{-1}(t) - F_Y^{-1}(t))^2 \, dt,$$

$\mu = \text{law}(X), \nu = \text{law}(Y)$, and, for $z \in \mathbb{R}$, $(z)_+^2 = (0 \vee z)^2$ denotes the squared hinge loss. Here, the numerator captures the degree to which $X$ fails to dominate $Y$ whereas the denominator serves as a normalizing constant so that $\varepsilon_{\mathcal{W}_2(\mu, \nu)} \in [0, 1]$. Indeed, as $(x - y)_+^2 + (y - x)_+^2 = (x - y)^2$, we see that $\varepsilon_{\mathcal{W}_2}(\mu, \nu) = 0$ precisely when $F_Y^{-1}(t) \leq F_X^{-1}(t)$ for a.e. $t \in (0, 1)$ whereas $\varepsilon_{\mathcal{W}_2}(\mu, \nu) = 1$ when the opposite inequality holds. del Barrio et al. [2018] then propose to test the null hypothesis $\varepsilon_{\mathcal{W}_2}(\mu, \nu) \leq \varepsilon_0$ for some $\varepsilon_0$ sufficiently close to 0, corresponding to the case where $X$ almost dominates $Y$ in the stochastic order, versus the alternative hypothesis $\varepsilon_{\mathcal{W}_2}(\mu, \nu) > \varepsilon_0$. To this end, they provide a central limit theorem for the statistic $\varepsilon_{\mathcal{W}_2}$ and propose to construct the rejection region via bootstrap estimation of the limiting variance. Similar results were obtained for a notion of almost second order stochastic dominance in Nitsure et al. [2023].

We highlight that the index $\varepsilon_{\mathcal{W}_2}$ can be connected to OT. Indeed, by Theorem 2.9 in Santambrogio [2015], the numerator and the denominator in $\varepsilon_{\mathcal{W}_2}(\mu, \nu)$ can be written, respectively, as $\inf_{\pi \in \Pi(\mu,\nu)} \int (y-x)_+^2 d\pi(x,y)$, and $\inf_{\pi \in \Pi(\mu,\nu)} \int (y-x)^2 d\pi(x,y)$, where $\Pi(\mu,\nu)$ denotes the set of all couplings of $(\mu,\nu)$. These problems are (univariate) instances of the well-studied OT problem

$$\mathsf{OT}_c(\mu,\nu) := \inf_{\pi \in \Pi(\mu,\nu)} \int c \, d\pi, \tag{1}$$

where $c : \mathbb{R}^d \times \mathbb{R}^d \to \mathbb{R}_+$ is a given cost function. We highlight that the costs considered herein are such that an optimal coupling always exists (see Theorem 4.1 in Villani, 2009). This connection will serve as our basis for extending the notion of almost stochastic dominance to the multivariate setting.

## 2.2 Multivariate FSD and its Relaxation via Optimal Transport with Compatible Costs

In the sequel, we provide a general framework for assessing multivariate almost FSD using a purely OT-based methodology. Following Chapter 6 and Theorem 6.B.1. in Shaked and Shanthikumar [2007], given the random vectors $X, Y \in \mathbb{R}^d$, we say that $X$ dominates $Y$ in the usual stochastic order (denoted $X \succeq_{\mathrm{FSD}} Y$) provided that there exists a coupling $(\hat{X}, \hat{Y})$ of $(X, Y)$ satisfying $\mathbb{P}(\hat{X} \geq \hat{Y}) = 1$ (i.e. for each $i = 1, \ldots, d$, $\hat{X}_i \geq \hat{Y}_i$ with probability one). This condition can be cast as follows:

**Lemma 1.** *Letting $\mu$ (resp. $\nu$) denote the law of $X$ (resp. $Y$), $X \succeq_{\mathrm{FSD}} Y$ if $\mathsf{OT}_c(\mu,\nu) = 0$, where $c : \mathbb{R}^d \times \mathbb{R}^d \to \mathbb{R}_+$ is the cost function $c(x,y) = \mathbb{1}_{\{x \leq y\}}(x,y)$.*

Evidently, the cost $\mathbb{1}_{\{x \leq y\}}(x,y)$ in Lemma 1 can be replaced by any nonnegative cost function $c(x,y)$ satisfying $c(x,y) = 0$ if and only if $x \leq y$ and it still holds that $X \succeq_{\mathrm{FSD}} Y$ if $\mathsf{OT}_c(\mu,\nu) = 0$. We denote the set of all such costs by $\mathfrak{C}_\leq$.

**Definition 1** (OT Costs Compatible with Multivariate FSD). *The set of all cost functions which are compatible with multivatiate FSD in the sense that $\mathsf{OT}_c(\mathrm{law}(X), \mathrm{law}(Y)) = 0$ implies that $X \succeq_{\mathrm{FSD}} Y$ is given by $\mathfrak{C}_\leq = \{c : \mathbb{R}^d \times \mathbb{R}^d \to \mathbb{R}_+ \text{ such that } c(x,y) = 0 \text{ if and only if } x \leq y\}$.*

A simple recipe for generating cost functions in $\mathfrak{C}_\leq$ is to take any univariate function $h : \mathbb{R} \to \mathbb{R}_+$ with the property that $h^{-1}(\{0\}) = (-\infty, 0]$ and define $c(x,y) = \sum_{i=1}^d h(y_i - x_i)$. For notational simplicty, we write $\mathsf{OT}_h$ to denote the OT cost with this type of cost function even when the aforementioned property does not hold. The results presented in the following sections require some additional smoothness assumptions on the function $h$ discussed above which we summarize presently.

**Definition 2** (Smooth Costs). *The function $h : \mathbb{R} \to \mathbb{R}_+$ satisfies the smoothness condition $(\mathrm{SC_d})$ if $h$ is Lipschitz continuous with constant $L \geq 0$ (that is, $|h(x) - h(y)| \leq L|x - y|$ for every $x, y \in \mathbb{R}$) and, for $k = \lfloor d/2 \rfloor + 1$, $h$ is $k$-times continuously differentiable with derivatives of order $s \leq k$ satisfying $|h^{(s)}(x)| \leq C_k(1 + |x|)^{p_k}$ for some $C_k < \infty$ and $p_k > 1$ which may depend on $k$.*

We now discuss some examples of cost functions of interest.

**Example 1** (Examples of OT Costs). *1) The function $h(z) = e^{-1/z}$ for $z \in (0, \infty)$ and $0$ otherwise is known to be smooth (see Example 1.3 in Tu, 2011) and satisfies $h^{-1}(\{0\}) = (-\infty, 0]$. It is easy to see that all derivatives of $h$ are $0$ on $(-\infty, 0]$, and decay to $0$ at infinity (and hence are bounded on $\mathbb{R}$) so that $h$ satisfies $(\mathrm{SC_d})$ for any $d \in \mathbb{N}$, and induces a cost function in $\mathfrak{C}_\leq$.*

*2) The squared hinge function $h(z) = (z)_+^2$ considered in Section 2.1 has linear growth, but is non-smooth. Although this function can be smoothed using e.g. mollification as introduced in Friedrichs [1944], this will result in a cost $c$ which is not an element of $\mathfrak{C}_\leq$ and may be costly to implement due to the convolution operation used in mollification.*

*3) The logistic function $h(z) = \log(1 + e^{\beta z})$ for $\beta > 0$ has linear growth and derivative $h'(z) = \beta \frac{e^{\beta z}}{1 + e^{\beta z}} = \beta \varsigma(\beta z)$, where $\varsigma(z) = \frac{1}{1 + e^{-z}}$ is the sigmoid function. As $\varsigma'(z) = \varsigma(z)(1 - \varsigma(z))$, it is easy to see that all derivatives of $h$ are bounded on $\mathbb{R}$ so that the assumption $(\mathrm{SC_d})$ is satisfied for any $d \in \mathbb{N}$. Although $h$ does not induce a cost in $\mathfrak{C}_\leq$, it is increasing and decays to $0$ faster than $e^{\beta z}$ as $z \to -\infty$. Moreover, if the induced cost satisfies $c(x,y) \leq \varepsilon_0$, then $\max_{i=1}^d(y_i - x_i) \leq \frac{1}{\beta} \log(e^{\varepsilon_0} - 1)$. Thus, if $\mathsf{OT}_c(\mu,\nu) = \int_{\{c \leq \varepsilon_0\}} c \, d\pi^\star + \int_{\{c > \varepsilon_0\}} c \, d\pi^\star = \varepsilon$ for an optimal plan $\pi^\star$ and*

*some small $\varepsilon > 0$, one has that $\int_{\{c > \varepsilon_0\}} c d\pi^\star \leq \varepsilon$ so that $\pi^\star(\{c > \varepsilon_0\}) \leq \varepsilon/\varepsilon_0$ i.e. $\pi^\star$ assigns at least mass $1 - \varepsilon/\varepsilon_0$ to points $(x,y)$ for which $\max_{i=1}^d (y_i - x_i) \leq \frac{1}{\beta} \log(e^{\varepsilon_0} - 1)$. Hence, for large $\beta$, $c$ enforces similar properties to a cost in $\mathfrak{C}_+$. Note also that $h$ can be viewed as a smooth surrogate for the $0/1$ loss for large $\beta$.*

At this point, a multivariate analogue to the univariate almost stochastic domination could be defined by analogy with Section 2.1. However, we highlight two major impasses which make the entropically regularized index considered in the following sections a far more palatable option in dimension $d > 1$. First, it is well-known that the expected rate of convergence of empirical OT generally suffers from the curse of dimensionality in statistical estimation; scaling as $n^{-1/d}$ (cf. e.g. Manole and Niles-Weed, 2024). Although Hundrieser et al. [2022] improve these rates as to depend on the minimum of the intrinsic dimensions of $\mu, \nu$ in place of $d$, entropic optimal transport exhibits a preferable parametric rate of convergence. Next, solving the OT problem numerically between two finitely discrete distributions supported on $N$ points requires solving a linear program in $N^2$ variables which can be prohibitive for even moderately sized problems.

## 3 Entropic Regularization of OT with Multivariate FSD Compatible Costs

Before defining the regularized index, we first provide some background on entropic optimal transport (EOT) with a cost $c : \mathbb{R}^d \times \mathbb{R}^d \to \mathbb{R}_+$. EOT is defined by regularizing the OT problem (1) as

$$\mathsf{OT}_{c,\lambda}(\mu, \nu) = \inf_{\pi \in \Pi(\mu, \nu)} \int c d\pi + \lambda \mathsf{D}_{\mathsf{KL}}(\pi || \mu \otimes \nu), \tag{2}$$

where $\lambda \geq 0$ is a regularization parameter and $\mathsf{D}_{\mathsf{KL}}$ is the Kullback-Leibler divergence defined by $\mathsf{D}_{\mathsf{KL}}(\rho, \eta) = \int \log\left(\frac{d\rho}{d\eta}\right) d\rho$ if $\rho$ is absolutely continuous with respect to $\eta$ and $\mathsf{D}_{\mathsf{KL}}(\rho, \eta) = +\infty$ otherwise. When $\lambda = 0$, we recover the standard OT problem. If $c \in L^1(\mu \otimes \nu)$, (2) admits a unique solution and is paired in strong duality with the problem

$$\sup_{\varphi \in L^1(\mu), \psi \in L^1(\nu)} \int \varphi d\mu + \int \psi d\nu - \lambda \int e^{\frac{\varphi(x) + \psi(y) - c(x,y)}{\lambda}} d\mu \otimes \nu(x,y) + \lambda. \tag{3}$$

Solutions to (3) are known to be almost surely unique up to additive constants (i.e. if $(\varphi, \psi), (\varphi', \psi')$ solve (3), $\varphi = \varphi' + C$ $\mu$-almost surely and $\psi = \psi' - C$ $\nu$-almost surely for some constant $C \in \mathbb{R}$) and are uniquely determined for $\mu$-a.e. $x$ and $\nu$-a.e. $y$ by the so-called Schrödinger system

$$e^{-\varphi(x)/\lambda} = \int e^{\frac{\psi(y) - c(x,y)}{\lambda}} d\nu(y), \quad e^{-\psi(y)/\lambda} = \int e^{\frac{\varphi(x) - c(x,y)}{\lambda}} d\mu(x), \tag{4}$$

which implies that $\int e^{\frac{\varphi(x) + \psi(y) - c(x,y)}{\lambda}} d\mu \otimes \nu(x,y) = 1$. EOT potentials satisfying (4) on the whole space are known to exist and are unique up to additive constants (see Lemma 6). We refer the reader to Nutz [2021] for a comprehensive introduction on EOT including all stated results.

Entropic regularization of OT problems was introduced in the seminal work of Cuturi [2013] as a means to accelerate computation by utilizing Sinkhorn-Knopp's matrix scaling algorithm which can be efficiently implemented on GPUs. Interestingly, entropic regularization also alleviates the curse of dimensionality rates in statistical estimation inherent to standard OT; for instance Genevay et al. [2019] and Mena and Niles-Weed [2019] show that the plug-in estimator for the EOT cost with fixed $\lambda > 0$ and cost $c(x,y) = \|x - y\|^2$ achieves a parametric expected rate of convergence with a dimension dependent constant (see also Groppe and Hundrieser, 2023, Stromme, 2023 for related results with the dimension replaced by the minimum intrinsic dimension of $\mu$ and $\nu$); del Barrio et al. [2023] and Goldfeld et al. [2024b] further establish a central limit theorem in this setting.

Given that EOT is meant to approximate OT, we derive the properties of $\mathsf{OT}_{h,\lambda}(\mu, \nu)$ and its solutions as $\lambda \downarrow 0$. First, we quantify the rate of convergence of $\mathsf{OT}_{h,\lambda}$ to $\mathsf{OT}_{h,0}$ under mild conditions, then show that solutions of $\mathsf{OT}_{h,\lambda}$ converge to solutions of $\mathsf{OT}_{h,0}$ as $\lambda \downarrow 0$ in a suitable sense.

**Theorem 1** (Stability as $\lambda \downarrow 0$)**.** *Let $\mu, \nu \in \mathcal{P}(\mathbb{R}^d)$ have finite first moment, $h$ be a function satisfying $(\mathsf{SC_d})$, and fix $\delta > 0$. Then,*

*1. for any $\lambda \in [0, 1)$ satisfying $\lfloor \lambda^{-d} \rfloor \geq (20/\delta)^d$, we have that*

$$0 \leq \mathsf{OT}_{h,\lambda}(\mu, \nu) - \mathsf{OT}_{h,0}(\mu, \nu) \leq d\lambda \log(1/\lambda) + \frac{4L\lambda}{\delta}(5C_\delta + 20d)$$

*for $C_\delta = \sum_{j=1}^d \int |x_j|^{1+\delta} d\mu_0(x) \wedge \sum_{j=1}^d \int |x_j|^{1+\delta} d\mu_1(x)$.*

*2. if $\pi_\lambda$ is the unique solution to $\mathsf{OT}_{h,\lambda}(\mu, \nu)$ for $\lambda > 0$, then there exists a subsequence of $(\pi_\lambda)_{\lambda \downarrow 0}$ which converges weakly to a solution of $\mathsf{OT}_{h,0}(\mu, \nu)$.*

We highlight that the implications of Theorem 1 require only the Lipschitz condition imposed in $(SC_d)$. The proof of the first result follows that of Theorem 3.3 in Eckstein and Nutz [2023] which controls the error of approximating the OT cost and OT plan using discretizations of the measures at play. The main novelty in our approach is to provide explicit constants and a simple argument showing that the rate at which a general measure on $\mathbb{R}^d$ can be approximated by a finitely discrete measure on at most $n$ points under the 1-Wasserstein distance scales at worst as $n^{-1/d}$ for sufficiently large $n$. The second statement is proved using the machinery of $\Gamma$-convergence (see Maso, 1993). Complete proofs are provided in Appendix C.2.

## 4 Entropic Multivariate FSD Violation Ratio and Testing

By analogy with the univariate case described in Section 2.1, we consider a normalized index of stochastic order violation given by

$$\varepsilon_{h,\lambda}(\mu, \nu) = \frac{\mathsf{OT}_{h,\lambda}(\mu, \nu)}{\mathsf{OT}_{\bar{h},\lambda}(\mu, \nu)}, \tag{5}$$

we adopt the convention that $\varepsilon_{h,\lambda}(\mu, \nu) = 0$ whenever $\mathsf{OT}_{\bar{h},\lambda}(\mu, \nu) = 0$. Here, $\mathsf{OT}_{\bar{h},\lambda}(\mu, \nu)$ is the EOT problem with cost $c(x, y) = \sum_{i=1}^d h(y_i - x_i) + h(x_i - y_i)$. This cost function is induced by the function $\bar{h}(z) = h(z) + h(-z)$ and hence satisfies $(SC_d)$ provided that $h$ satisfies $(SC_d)$. Moreover, $\mathsf{OT}_{h,\lambda}(\mu, \nu) \leq \mathsf{OT}_{\bar{h},\lambda}(\mu, \nu)$ by construction so that $\varepsilon_{h,\lambda}(\mu, \nu) \in [0, 1]$ yielding a normalized index. The corresponding notion of entropic multivariate almost stochastic dominance can thus be defined.

**Definition 3** (Entropic Multivariate Almost FSD). *We define $(h, \lambda, \varepsilon_0) - FSD$, the entropic multivariate almost FSD via the violation ratio as follows:*

$$\mu \underset{(h,\lambda,\varepsilon_0)-FSD}{\succeq} \nu \text{ if } \varepsilon_{h,\lambda}(\mu, \nu) \leq \varepsilon_0.$$

In light of Theorem 1, $\lim_{\lambda \downarrow 0} \varepsilon_{h,\lambda}(\mu, \nu) = \frac{\mathsf{OT}_{h,0}(\mu,\nu)}{\mathsf{OT}_{\bar{h},0}(\mu,\nu)}$ with the convention that this latter quantity is zero when $\mathsf{OT}_{\bar{h},0}(\mu, \nu) = 0$. In the case that $h$ is the squared hinge function described in Example 1 and $d = 1$ we recover the univariate index from Section 2.1. As aforementioned, the squared hinge function is not sufficiently smooth to enable us to characterize the asymptotic fluctuations of the empirical index in arbitrary dimensions. See Example 1 which lists other examples of costs satisfying $(SC_d)$; this condition is sufficient for the following statistical developments. We underscore that the choice of cost influences the notion of stochastic dominance reflected by the violation ratio. In applications where a practitioner wishes to formulate a domain-specific notion of dominance, a data-driven approach can be employed by replacing the fixed cost $h$ in the previous development by a collection of costs $(h_{i,\theta_i})_{i=1}^d$ for each dimension where $\theta_i$ is the corresponding parameter (e.g. $\beta$ in the logistic function) and optimizing over these parameters. The results presented herein readily adapt to this setting, but, for simplicity, we restrict our attention to the case of a fixed $h$ throughout.

### 4.1 Statistical Properties

We now lay the groundwork for performing principled statistical inference with the empirical estimator of the entropic index $\varepsilon_{h,\lambda}$. Namely, we establish the asymptotic properties of the plug-in estimator $\varepsilon_{h,\lambda}(\hat{\mu}_n, \hat{\nu}_n)$, where $\hat{\mu}_n = \frac{1}{n}\sum_{i=1}^n \delta_{X_i}, \hat{\nu}_n = \frac{1}{n}\sum_{j=1}^n \delta_{Y_j}$ are the empirical distributions from $n$ independent observations, $(X_i)_{i=1}^n$ and $(Y_j)_{j=1}^n$ of $\mu$ and $\nu$ respectively. Furthermore, we establish consistency of the bootstrap procedure. To this end, given sets of $n$ iterations observations of $\mu$ and $\nu$, $(X_i)_{i=1}^n$ and $(Y_j)_{j=1}^n$ as above and sets $(X_i^B)_{i=1}^n$ and $(Y_j^B)_{j=1}^n$ of $n$ independent samples from

$\hat{\mu}_n$ and $\hat{\nu}_n$, $\hat{\mu}_n^B := \frac{1}{n} \sum_{i=1}^{n} \delta_{X_i^B}$ and $\hat{\nu}_n^B := \frac{1}{n} \sum_{j=1}^{n} \delta_{Y_j^B}$ are the corresponding bootstrap empirical distributions. $\mathbb{P}^B$ denotes the conditional probability given the data.

**Theorem 2** (Limit distribution and bootstrapping). *Assume that $\mu, \nu \in \mathcal{P}(\mathbb{R}^d)$ are sub-Gaussian with a shared parameter $\tau^2 > 0$ and that $h$ satisfies $(\mathrm{SC_d})$. Let $(\varphi_h, \psi_h)$ and $(\varphi_{\bar{h}}, \psi_{\bar{h}})$ be any pairs of optimal potentials for $\mathsf{OT}_{h,\lambda}(\mu, \nu)$ and $\mathsf{OT}_{\bar{h},\lambda}(\mu, \nu)$ respectively satisfying the Schrödinger system (4) on $\mathbb{R}^d \times \mathbb{R}^d$. Then, if $\mathsf{OT}_{\bar{h},\lambda}(\mu, \nu) > 0$,*

1. *$\sqrt{n} \left( \varepsilon_{h,\lambda}(\hat{\mu}_n, \hat{\nu}_n) - \varepsilon_{h,\lambda}(\mu, \nu) \right) \xrightarrow{d} N(0, \sigma^2)$, a mean-zero Gaussian with variance $\sigma^2 = \mathrm{var}_\mu \left( \frac{1}{\mathsf{OT}_{\bar{h},\lambda}(\mu,\nu)} \varphi_h - \frac{\mathsf{OT}_{h,\lambda}(\mu,\nu)}{\mathsf{OT}_{\bar{h},\lambda}(\mu,\nu)^2} \varphi_{\bar{h}} \right) + \mathrm{var}_\nu \left( \frac{1}{\mathsf{OT}_{\bar{h},\lambda}(\mu,\nu)} \psi_h - \frac{\mathsf{OT}_{h,\lambda}(\mu,\nu)}{\mathsf{OT}_{\bar{h},\lambda}(\mu,\nu)^2} \psi_{\bar{h}} \right).$*

2. *If $\sigma^2 > 0$, $\sup_{t \in \mathbb{R}} \left| \mathbb{P}^B \left( \sqrt{n}(\varepsilon_{h,\lambda}(\hat{\mu}_n^B, \hat{\nu}_n^B) - \varepsilon_{h,\lambda}(\hat{\mu}_n, \hat{\nu}_n)) \leq t \right) - \mathbb{P}(N(0, \sigma^2) \leq t) \right| \xrightarrow{\mathbb{P}} 0.$*

Observe that the limiting distribution in Theorem 2 is non-pivotal in the sense that the variance depends on the population distributions $(\mu, \nu)$ rendering direct estimation of the limiting variance highly non-trivial. The bootstrap consistency result in the second point enables us to establish confidence intervals for $\varepsilon_{h,\lambda}(\mu, \nu)$. Explicitly, if $\zeta_\beta$ denotes the smallest value of $t \in \mathbb{R}$ for which $\mathbb{P}^B \left( \varepsilon_{h,\lambda}(\hat{\mu}_n^B, \hat{\nu}_n^B) \leq t \right) \geq 1 - \beta$ for any $\beta \in (0, 1)$, then $\varepsilon_{h,\lambda}(\mu, \nu) \in [0, 2\varepsilon_{h,\lambda}(\hat{\mu}_n, \hat{\nu}_n) - \zeta_\alpha]$ with probability approaching $1 - \alpha$ for any $\alpha \in (0, 1)$ due to Lemma 23.3 in Van der Vaart [2000].

The proof of Theorem 2 is based on the functional delta method [Römisch, 2006] which extends the standard delta method to functionals defined on normed vector spaces, following the framework of Goldfeld et al. [2024b]. Formally, this approach consists of showing that the functional mapping $\tau^2$-sub-Gaussian distributions $(\eta, \rho)$ to $\varepsilon_{h,\lambda}(\eta, \rho)$ is directionally differentiable at $(\mu, \nu)$ and Lipschitz continuous in a suitable sense and that the relevant potentials lie in a space of sufficiently smooth functions using the assumption $(\mathrm{SC_d})$. Smoothness of the potentials is crucial to ensure that the empirical processes $\sqrt{n}(\hat{\mu}_n - \mu), \sqrt{n}(\hat{\nu}_n - \nu)$ converge when treated as functionals on the aforementioned space of smooth functions. Complete details are included in Appendix C.3, and a primer on the functional delta method can be found in Appendix D.

**Remark 1** (On Theorem 2). *1) We note that the condition that $\mathsf{OT}_{\bar{h},\lambda}(\mu, \nu) > 0$ in Theorem 2 is satisfied except in certain degenerate settings. Indeed, for a general nonnegative cost $c$, $\mathsf{OT}_{c,\lambda}(\mu, \nu) = 0$ if and only if $\int c \, d\mu \otimes \nu = 0$ as follows from the fact that $\mathsf{D_{KL}}(\pi \| \mu \otimes \nu) \geq 0$ with equality if and only if $\pi = \mu \otimes \nu$. In particular, $\mathsf{OT}_{\bar{h},\lambda}(\mu, \nu) = 0$ if and only if $h(x_i - y_i) = h(y_i - x_i) = 0$ for every $x \in \mathrm{spt}(\mu)$ and $y \in \mathrm{spt}(\nu)$ and every $i \in \{1, \ldots, d\}$. If $h$ is chosen as to generate a cost function which is compatible with multivariate FSD (recall Definition 1), $h^{-1}(\{0\}) = (-\infty, 0]$ so that $\mathsf{OT}_{\bar{h},\lambda}(\mu, \nu) = 0$ if and only if $\mu$ and $\nu$ are point masses at some shared $a \in \mathbb{R}^d$.*

*2) Theorem 2 is presented in the balanced case with empirical measures from $n$ samples. In the case where $\hat{\mu}_n$ and $\hat{\nu}_m$ are empirical measures from $n \neq m$ samples with $\frac{n}{n+m} \to s \in (0, 1)$, the implications of Theorem 2 are easily seen to hold with $\sqrt{\frac{nm}{n+m}}$ in place of $\sqrt{n}$ and $\sigma_s^2 = \mathrm{var}_\mu \left( \frac{1-s}{\mathsf{OT}_{\bar{h},\lambda}(\mu,\nu)} \varphi_h - \frac{s\mathsf{OT}_{h,\lambda}(\mu,\nu)}{\mathsf{OT}_{\bar{h},\lambda}(\mu,\nu)^2} \varphi_{\bar{h}} \right) + \mathrm{var}_\nu \left( \frac{1-s}{\mathsf{OT}_{\bar{h},\lambda}(\mu,\nu)} \psi_h - \frac{s\mathsf{OT}_{h,\lambda}(\mu,\nu)}{\mathsf{OT}_{\bar{h},\lambda}(\mu,\nu)^2} \psi_{\bar{h}} \right)$ in place of $\sigma^2$.*

### 4.2 Multivariate FSD Hypothesis Testing In ML Models Benchmarking

With the statistical properties of the violation ratio in hand, we now consider using the violation ratio in the context of statistical testing for $(h, \lambda, \varepsilon_0)-$ FSD. Consider two $d$-dimensional distributions $\mu, \nu$. In our application, these will correspond to the distributions of performance of two language models' responses evaluated on $d$ metrics. Given $n, m$ samples from $\mu, \nu$ respectively, we can apply Theorem 2 to create statistically valid tests comparing $\mu$ and $\nu$. Similarly to Nitsure et al. [2023], we consider both absolute and relative testing (see Nitsure et al., 2023 for a complete discussion).

**Absolute testing** The most straightforward application of Theorem 2 is to specify a desired threshold $\varepsilon_0$ and consider the following hypothesis test for $(h, \lambda, \varepsilon_0)-$ FSD: $H_0 : \mu \underset{(h,\lambda,\varepsilon_0)-\mathrm{FSD}}{\not\succeq} \nu$ versus the alternative $H_1 : \mu \underset{(h,\lambda,\varepsilon_0)-\mathrm{FSD}}{\succeq} \nu$. Note that $\nu$ dominating $\mu$ would be tested separately. Given a desired

confidence $1 - \alpha$, the central limit theorem and bootstrap results in Theorem 2 suggest rejecting $H_0$ if

$$\varepsilon_{h,\lambda}(\hat{\mu}_n, \hat{\nu}_m) \leq \varepsilon_0 + \sqrt{\frac{m+n}{mn}} \sigma_B(\hat{\mu}_n, \hat{\nu}_m) \Phi^{-1}(\alpha), \tag{6}$$

where $\sigma_B^2$ is the bootstrapped variance (See Algorithm 1) and $\Phi$ is the CDF of the standard normal distribution. By Theorem 2, this test will be asymptotically valid.

**Relative testing** A downside of absolute testing is that it requires specifying a threshold $\varepsilon_0$. This threshold can be meaningful in pairwise comparisons, but when multiple distributions (e.g. multiple language models evaluations) are being compared for ranking purposes, it is difficult to determine a priori what threshold to use to ensure that the distributions can be separated. As in Nitsure et al. [2023], we therefore also present a *relative* test that compares each of $k$ random vectors with measures $\mu_1, \ldots, \mu_k$ using a one-versus-all violation ratio. First, consider all pairs of violations ratios between the $k$ measures: $\varepsilon_{ij}^{(h,\lambda)} = \varepsilon_{h,\lambda}(\mu_i, \mu_j)$ for $i, j \in \{1 \ldots k\}, i \neq j$. Let $M = (\mu_1, \ldots \mu_k)$, and define the one-versus-all violation ratio of the dominance of $\mu_i$ on all other variables $\mu_j, j \neq i$: $\varepsilon_i^{(h,\lambda)}(M) = \frac{1}{k-1} \sum_{j \neq i} \varepsilon_{ij}^{(h,\lambda)}$. We can then define *relative stochastic dominance* $(h,\lambda)$-R-FSD as $\mu_{i_1} \underset{R-\text{FSD}}{\succeq} \mu_{i_2} \cdots \underset{R-\text{FSD}}{\succeq} \mu_{i_k} \iff \varepsilon_{i_1}^{(h,\lambda)}(M) \leq \cdots \leq \varepsilon_{i_k}^{(h,\lambda)}(M)$. Here the most dominating model is the one with the lowest one-versus-all violation ratio. Testing for relative dominance of $\mu_i$ on $\mu_j$ we can then compare their one-versus-all ratios via the following statistic: $\Delta \varepsilon_{ij}^{(\ell)}(M) = \varepsilon_i^{(\ell)}(M) - \varepsilon_j^{(\ell)}(M)$. To test for $(h,\lambda)$-R-FSD of $\mu_i$ versus $\mu_j$ then, we have the null hypothesis $H_0 : \Delta \varepsilon_{ij}(M) \geq 0$ versus the alternative $H_1 : \Delta \varepsilon_{ij}(M) < 0$. It is possible to extend the central limit theorem and bootstrapping results in Theorem 2 to this relative statistic under an independence assumption (omitted for brevity). Let $\hat{M}_n = (\hat{\mu}_{1,n}, \ldots \hat{\mu}_{k,n})$ be the empirical measures for $n$ samples from each distribution. As in the absolute case, we then reject $H_0$ with a confidence $1 - \alpha$ if: $\Delta \varepsilon_{i_1, i_2}(\hat{M}_n) \leq \sqrt{\frac{1}{n}} \sigma_{B,\text{relative}}(i_1, i_2) \Phi^{-1}(\alpha)$ where $\sigma_{B,\text{relative}}^2(i_1, i_2)$ is the bootstrapped variance (see Algorithm 1 for the variance expression).

**Multitesting and Ranking** To apply the above multivariate FSD violation ratio hypothesis tests to ranking of multiple distributions, we follow the approach of Nitsure et al. [2023]. We aggregate the set of all pairwise tests, ensure multitesting statistical validity via Family-Wise Error Rate (FWER) control, and, if needed, aggregate the pairwise test results into a numerical ranking. Our approach is described below and summarized in Algorithm 1. The overall complexity of performing one pairwise test is dominated by the cost of computing the EOT cost with $h$ and $\tilde{h}$. To this end, the Sinkhorn algorithm is used [Cuturi, 2013], which computes EOT between distributions on $N$ points with a complexity of $O(N^2 K(d) + N^2)$, where $K(d)$ denotes the cost of complexity of computing the cost $c(x, y)$ between $x, y \in \mathbb{R}^d$.

**Ranking multiple distributions** In our experiments below, we seek to use the pairwise tests above to obtain a statistically valid ranking of a set of $k$ random vectors $X^{(i)}$ with measures $\mu^{(i)}$, e.g. samples of $X^{(i)}$ can be the per-sample evaluation metrics for a set of language models. To rank $k$ models at a specified significance level $\alpha$, we first test all $k^2 - k$ pairs $(\mu^{(i)}, \mu^{(j)})$, $i \neq j$, employing a FWER correction (see next paragraph) to guarantee a valid control on the overall false rejection rate. This yields a set of trinary outcomes for each $(i, j)$ pair, with 1 if the null is rejected in the positive direction, -1 if the null is rejected in the negative direction, and 0 if the null is not rejected. These pairwise rankings are then combined into a single rank using a simple Borda count [de Borda, 1781] rank aggregation algorithm.

**Multitesting FWER control** When running a family of $T$ tests each at a significance level $1 - \alpha$, the true probability that at least one test falsely rejects the null scales with $T$. If the output of all $T$ tests needs to be trusted simultaneously, instead it is desirable that the probability of *any* test falsely rejects the null is less than or equal to the specified $\alpha$. Achieving this requires adjusting, or "correcting" the significance level of each of the $T$ tests, in a process called Family-Wise Error Rate (FWER) control. In the present work, we use the Bonferroni correction, which sets the significance level of the $i$th test to $1 - \alpha_i$ with $\alpha_i = \alpha/T$. Note that while the Bonferroni correction is known to be pessimistic, we choose it as it sets uniform significance levels for all tests, as opposed to other strategies such as the Holm correction [Holm, 1979] which is tighter but yields highly nonuniform statistical power across the family of tests. Exploring sensible ways to employ nonuniform FWER control approaches in the context of performance ranking is an interesting avenue for future work.

# 5 Experiments

All experiments were run on NVIDIA A100 80GB GPUs using PyTorch [Ansel et al., 2024] (v.2.3.0, BSD-3 license) and the Python Optimal Transport package [Flamary et al., 2021] (v.0.9.3, MIT license) to compute optimal transport distances with and without regularization. Code for these experiments is available at https://github.com/IBM/stochastic-order-eval.

## 5.1 Synthetic Data Experiment

In this section we analyze our method on a synthetic toy dataset that enables us to parametrically control the level of the multivariate stochastic dominance between two random variables. Given a dimension $d$, a parameter $p \in [0, 1]$, and mean and variance parameters $\mu, \sigma^2$, our synthetic dataset is generated by sampling from the multivariate random variables $X, Y \in \mathbb{R}^d$:

- $X_i \sim \mathcal{N}(\mu, \sigma^2)$ for $i = 1, \ldots, d$
- $Y_i = X_i + (2 \cdot B_i(p) - 1)U_i$ with $B_i(p) = \text{Bernoulli}(p) \in \{0, 1\}$ and $U_i = \text{Uniform}(0, 1)$.

These variables $X$ and $Y$ are designed in such a way that $p$ parametrizes the dominance of $Y$ over $X$. In particular, $X \underset{(h,\lambda)-\text{FSD}}{\succcurlyeq} Y$ if $p < 0.5$, and $Y \underset{(h,\lambda)-\text{FSD}}{\succcurlyeq} X$ if $p > 0.5$.

As a baseline for our synthetic experiments, we also compute the violation ratio for the standard FSD framework using unregularized OT (EMD), i.e. $\varepsilon_{\text{hinge},0}$, as a function of $p$ for fixed $d = 5$, $\mu = 0$, $\sigma^2 = 1.0$ and $N = 100$ samples from $X$ and $Y$. We then investigate how well this baseline is approximated by $\varepsilon_{\log,\lambda>0}$, the entropically regularized ratio with a logistic cost as in Example 1. Fig. 1 shows that as the entropic regularization parameter $\lambda$ decreases towards 0 and as the gain of the logistic cost $\beta$ increases, $\varepsilon_{\log,\lambda>0}$ converges towards $\varepsilon_{\text{hinge},0}$ across all values of $p \in [0, 1]$. In all cases, multivariate FSD violation ratio predicts linearly $p$, indicating that it is captures well the FSD violations. This experiment indicates that, to best approximate the standard FSD violation ratio, $\lambda$ should be taken as small as possible (c.f. Theorem 1) and $\beta$ should be taken as large as possible (c.f. Example 1). There is, in practice, a tradeoff that must be made when computing the regularized index, as Sinkhorn's algorithm requires the matrix $e^{-C/\lambda}$, where $C$ is the matrix of pairwise costs. As such, if the ratio $\beta/\lambda$ is too large, numerical underflow will occur and the algorithm will fail. As a rule of thumb, it is recommended to set $\beta$ first and increase the value of $\lambda$ if instability occurs.

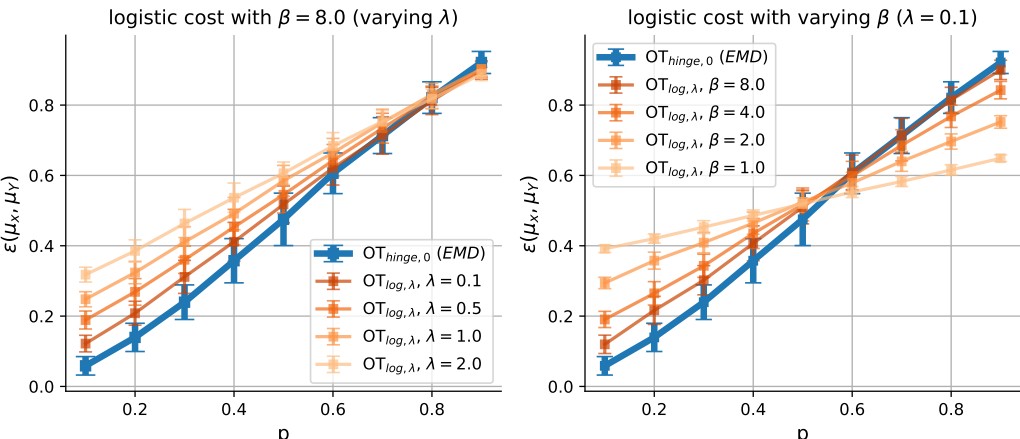

Figure 1: Convergence of $\varepsilon_{\log,\lambda>0}$ towards $\varepsilon_{\text{hinge},0}$ in the synthetic dataset introduced in this section. Left panel: for a fixed parameter $\beta = 8$ of the logistic cost, $\varepsilon_{\log,\lambda>0}$ converge towards $\varepsilon_{\text{hinge},0}$ as $\lambda$ is decreased toward 0. Right panel: for a fixed entropic regularization parameter $\lambda = 0.1$, $\varepsilon_{\log,\lambda}$ converges towards $\varepsilon_{\text{hinge},0}$ as the gain of the logistic cost $\beta$ increases. All simulations were generated for $d = 5$, $\mu = 0$, $\sigma^2 = 1.0$ and $N = 100$. Points and error bars indicate average and standard deviation across 100 repetitions.

We now assess the power of our proposed test with as a function of the number of samples and the dimension. We consider the same setup as the previous experiment and set $p = 0.65$ so that

$Y \underset{(h,\lambda,0.5)-\mathrm{FSD}}{\succcurlyeq} X$. We then estimate the type I and type II error of the relative test statistic by averaging across 100 repetitions, these results are compiled in Fig. 2.

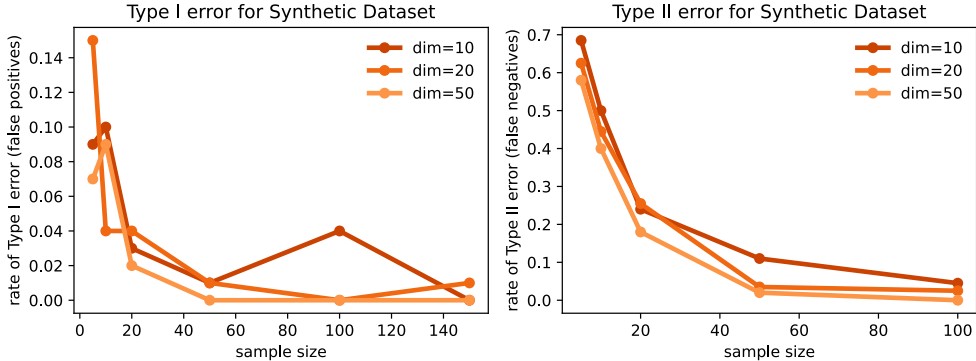

Figure 2: Type I and type II error of the relative test statistic as a function of the sample size $n$ in dimension $d \in \{10, 20, 50\}$. Here, $\beta = 8$ and $\lambda = 0.01d$.

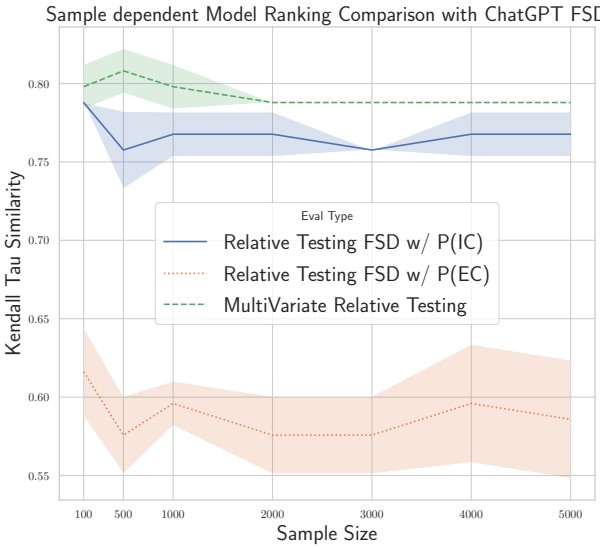

Figure 3: Mix Instruct Results: Comparison of Multivariate FSD to Reduction to univariate FSD with aggregation across the dimensions.

### 5.2 LLM Benchmarking

To test our method on real world scenarios, we have chosen a current topic of significant interest to the community: LLM Benchmarking. We show through our experiments that our method can provide a more holistic ranking of LLMs evaluated on different metrics as opposed to present strategies which involve mean win rate. To demonstrate our method's application to LLM Benchmarking, we have conducted assessments on two different sets of data.

**Mixinstruct** For our first evaluation we use the dataset from Jiang et al. [2023] (MIT license) that consists of responses from 12 different instruction following LLMs, with each response evaluated on 9 metrics such as BLEU, ROUGE, BERTScore, BARTScore, etc. The data has a train (100K rows) and test (5k rows) split where each row consists of an instruction, input sentence, the expected output from users, as well as the responses of a set of different LLMs with their decoding parameters and evaluation scores on different metrics. However for the test set, Jiang et al. [2023] also did a

pairwise evaluation of the responses from the models by asking ChatGPT which response was better. We use this test set and generate a ranking of the LLMs using Entropic Multivariate FSD (Algorithm 1), where for each LLM we construct an empirical measure on $\mathbb{R}^9$ using $n$ samples varying from 100 to 5000. We then compute the pairwise ratios for these empirical distributions using the logistic loss with $\beta = 0.2$, the regularization parameter $\lambda = 0.1$, and utilize the relative testing procedure from Section 4.2 to rank the 12 LLMS (see Fig. 4 for the ranking obtained with $n = 5000$ in appendix Appendix B). The confidence intervals are then generated using 1000 bootstrap repetitions. Finally, we compare the resulting ranking with the (univariate ranking) provided by ChatGPT scoring (which serves as a human proxy) as a function of the sample size $n$ using Kendall Tau similarity. These results are presented in Fig. 3.

We then compare different methods that reduce multivariate ordering to univariate FSD via aggregation. The first method is a portfolio aggregation with Independent Copula P(IC) [Nitsure et al., 2023], where a dimension is normalized with a global univariate CDF across all models and a geometric mean is performed across all dimensions. A univariate FSD is then applied on the resulting univariate random variables. The second method, referred to as portfolio aggregation with Empirical Copula P(EC) [Ruschendorf, 1976, Ulan et al., 2021], estimates a global multivariate CDF across all models, and then assigns to each evaluation vector the value of its CDF. Similarly, a univariate FSD is applied on this one dimensional data.

**Results** We see from Fig. 3 that the multivariate FSD, is sample efficient and has the highest Kendall tau rank similarity with GPT score. We hypothesize that this thanks to its ability to capture dependencies between the metrics. The independent copula P(IC), ignores the dependencies and hence lags a little behind but is still sample efficient. Whilst the empirical copula P(EC) captures the dependencies, it suffers from the curse of dimension and is not sample efficient.

## 6 Conclusion

In this paper, we proposed entropic multivariate FSD violation ratio as a statistic for assessing multivariate first order dominance. We addressed the convergence of these ratios as the entropic regularization goes to zero and established a central limit theorem and bootstrap consistency for this statistic. These statistical properties were leveraged in a framework for multivariate FSD testing which was applied to multi-metrics benchmarking machine learning models, showing its benefits in capturing the metric dependencies. Casting testing for stochastic order as an optimal transport problem with a smooth cost and devising an entropic regularization to ensure beneficial statistical and computational properties is an interesting framework that we envision to be useful and versatile for other stochastic orders. For instance the $\mu$-first order dominance of Galichon and Henry [2012] uses optimal transport maps as multivariate quantiles [Carlier et al., 2014] and defines a $\mu$-stochastic dominance; our entropic violation ratio framework can be extended to that case and, upon proving central limit theorems on the OT potentials, will lead to similar central limit theorems to the one presented in this work. Similarly, for the multivariate Lorenz order [Fan et al., 2024] that is of interest when the agent making the choice is risk averse. The Lorenz order can be expressed in terms of optimal transport maps and can be extended to our statistical testing framework using the tools introduced in this paper. We leave these developments for future work.

## Acknowledgments and Disclosure of Funding

G. Rioux is partially supported by the NSERC postgraduate fellowship PGSD-567921-2022.

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

# A   Algorithm

Algorithm 1 describes our multitesting-based ranking procedure, for both the absolute and relative testing frameworks.

---

**Algorithm 1** Multivariate Stochastic Order Multi-testing (relative and **absolute**)

---

1: **Input:** $\mu_1, ..., \mu_k$, $k$ models we want to rank corresponding to empirical measure $p_1 = \frac{1}{n}\sum_{i=1}^{n}\delta_{x_i^1}, \ldots p_k = \frac{1}{n}\sum_{i=1}^{n}\delta_{x_i^k}$, **Threshold:** $\tau$.

2: **Input:** Desired $h, \lambda$, $B$ number of bootstraps, $m = K^2$ number of comparisons, significance level $\alpha$.

3: **Cache the bootstraps samples**

4: **for** $j = 1$ **to** $k$ **do**

5:     $p_j^0 \leftarrow p_j$

6:     **for** $b = 1$ **to** $B$ **do**

7:        $p_j^b \leftarrow$ RESAMPLEWITHREPLACEMENT$(p_j, n)$

8: **Compute all violation ratios**

9: **for** $b = 0$ **to** $B$ **do**

10:     **for** $i = 1$ **to** $k$ **do**

11:        **for** $j = 1$ **to** $k$ **do**

12:           **if** $i \neq j$ **then**

13:              Cost matrix $[C_h]_{k,l} \leftarrow h([p_i^b]_k - [p_j^b]_l)$. {Use $h(z) = \log(1 + e^{\beta z})$. $[p_i^b]_k$ is the $k$th sample in $p_i^b$.}

14:              $[C_{\bar{h}}]_{k,l} \leftarrow [C_h]_{k,l} + [C_h]_{l,k}$.

15:              $\Pi_{i,j,b} \leftarrow$ Sinkhorn$(C_h, \lambda, p_i^b, p_j^b)$, $\bar{\Pi}_{i,j,b} \leftarrow$ Sinkhorn$(C_{\bar{h}}, \lambda, p_i^b, p_j^b)$. {Sinkhorn alg. with costs $C_h$, $C_{\bar{h}}$ and entropic reg. $\lambda$.}

16:              $\mathsf{OT}_{h,\lambda}(p_i^b, p_j^b) \leftarrow$ Trace$(C_h^\top \Pi_{i,j,b}) + \lambda D_{\mathsf{KL}}(\bar{\Pi}_{i,j,b}||p_i^b \otimes p_j^b)$.

17:              $\mathsf{OT}_{\bar{h},\lambda}(p_i^b, p_j^b) \leftarrow$ Trace$(C_{\bar{h}}^\top \bar{\Pi}_{i,j,b}) + \lambda D_{\mathsf{KL}}(\bar{\Pi}_{i,j,b}||p_i^b \otimes p_j^b)$.

18:              $\varepsilon_{b,i,j} \leftarrow \varepsilon_{h,\lambda}(p_i^b, p_j^b) = \frac{\mathsf{OT}_{h,\lambda}(p_i^b, p_j^b)}{\mathsf{OT}_{\bar{h},\lambda}(p_i^b, p_j^b)}$ in (5).

19: $\varepsilon_{b,i,i} = 0, \forall\, b, i$

20: **Compute the sum statistics**

21: **for** $b = 0$ **to** $B$ **do**

22:     **for** $i = 1$ **to** $k$ **do**

23:        $\varepsilon_b^i \leftarrow \frac{1}{k-1}\sum_j \varepsilon_{b,i,j}$

24: **Compute the relative statistics**

25: $\Delta\varepsilon_b^{i,j} = \varepsilon_b^i - \varepsilon_b^j, \forall b, i, j$

26: **Compute the Bootstrap Variance**

27: **for** $i = 1$ **to** $k$ **do**

28:     **for** $j = 1$ **to** $k$ **do**

29:        $\sigma_{ij} = \sqrt{\frac{1}{B-1}\sum_{b=1}^{B}(\Delta\varepsilon_b^{i,j} - \text{MEAN}(\Delta\varepsilon_b^{i,j}, b))^2}$

30:        $\sigma_{ij}^{\text{abs}} = \sqrt{\frac{1}{B-1}\sum_{b=1}^{B}(\varepsilon_{b,i,j} - \text{MEAN}(\varepsilon_{b,i,j}, b))^2}$

31: **Compute the test**

32: $\text{Win}_{ij} = \text{Win}_{ij}^{\text{abs}} = 0$

33: **for** $i = 1$ **to** $k$ **do**

34:     **for** $j = 1$ **to** $k$ **do**

35:        **if** $i \neq j$ and $\Delta\varepsilon_0^{i,j} - \frac{1}{\sqrt{n}}\sigma_{ij}\Phi^{-1}(\alpha/k^2) \leq 0$ **then**

36:           $\text{Win}_{ij} = 1$ {with confidence level $1 - \alpha/k^2$}

37:        **if** $i \neq j$ and $\varepsilon_{0.i,j} - \frac{1}{\sqrt{n}}\sigma_{ij}^{\text{abs}}\Phi^{-1}(\alpha/k^2) \leq \tau$ **then**

38:           $\text{Win}_{ij}^{\text{abs}} = 1$ {with confidence level $1 - \alpha/k^2$}

     rank = BORDA(Win) {with confidence level $1 - \alpha$}

     rank$_{\text{abs}}$ = BORDA(Win$^{\text{abs}}$) {with confidence level $1 - \alpha$}

39: **Return** rank, rank$_{\text{abs}}$

---

| Model | one-versus-all violation ratio |
|---|---|
| oasst-sft-4-pythia-12b-epoch-3.5 | 0.463332 |
| chatglm-6b | 0.472356 |
| alpaca-native | 0.474452 |
| vicuna-13b-1.1 | 0.475832 |
| llama-7b-hf-baize-lora-bf16 | 0.490481 |
| moss-moon-003-sft | 0.499978 |
| koala-7B-HF | 0.506661 |
| mpt-7b-instruct | 0.508465 |
| dolly-v2-12b | 0.512816 |
| stablelm-tuned-alpha-7b | 0.521765 |
| mpt-7b | 0.522466 |
| flan-t5-xxl | 0.558775 |

Figure 4: This table ranks the 12 models tested in the LLM benchmarking experiment using $n = 5000$ samples according to their one-versus-all violation ratio, $\varepsilon_i^{(h,\lambda)}(M) = \frac{1}{k-1} \sum_{j \neq i} \varepsilon_{ij}^{(h,\lambda)}$ where $\varepsilon_{ij}^{(h,\lambda)}$, is the pairwise violation ration of model $i$ compared with model $j$ (lower is better).

## B  Ranking from the Mix Instruct experiment

## C  Proofs of main results

### C.1  Proof of Lemma 1

Observe that the set $\{(x,y) \in \mathbb{R}^d \times \mathbb{R}^d : x \leq y\}$ is closed, as any limit point, $(x,y)$, of a sequence $\{(x_n, y_n)\}_{n \in \mathbb{N}}$ satisfying $x_n \leq y_n$ is such that $x \leq y$ as the relevant inequalities are preserved in the limit. It follows that $\mathbb{1}_{\{x \leq y\}}(x,y)$ is a lower semicontinuous function and hence there exists a coupling $\bar{\pi} \in \Pi(\mu, \nu)$ for which $\inf_{\pi \in \Pi(\mu,\nu)} \int \mathbb{1}_{\{x \leq y\}}(x,y)d\pi(x,y) = \int \mathbb{1}_{\{x \leq y\}}(x,y)d\bar{\pi}(x,y)$ for any choice of $\mu, \nu \in \mathbb{R}^d$ (cf. e.g. Theorem 4.1 in Villani, 2009).

Assume that $\inf_{\pi \in \Pi(\mu,\nu)} \int \mathbb{1}_{\{x \leq y\}}(x,y)d\pi(x,y) = 0$ such that there exists a coupling $\bar{\pi}$ for which $0 = \int \mathbb{1}_{\{x \leq y\}}(x,y)d\bar{\pi}(x,y) = \mathbb{P}_{(\hat{X},\hat{Y})\sim\bar{\pi}}(\hat{X} \leq \hat{Y}) = 1 - \mathbb{P}_{(\hat{X},\hat{Y})\sim\bar{\pi}}(\hat{X} > \hat{Y})$ i.e. $\mathbb{P}_{(\hat{X},\hat{Y})\sim\bar{\pi}}(\hat{X} \geq \hat{Y}) \geq \mathbb{P}_{(\hat{X},\hat{Y})\sim\bar{\pi}}(\hat{X} > \hat{Y}) = 1$ which, in light of Theorem 6.B.1. in [Shaked and Shanthikumar, 2007], implies that $X \underset{\text{FSD}}{\succcurlyeq} Y$. $\square$

### C.2  Proof of Theorem 1

Throughout, we fix the metric $\mathsf{d} : ((x,y),(x',y')) \in \mathbb{R}^{2d} \times \mathbb{R}^{2d} \mapsto \|x - x'\|_1 + \|y - y'\|_1$ on $\mathbb{R}^{2d}$. We first show that costs induced by functions $h$ satisfying $(\text{SC}_\mathsf{d})$ are Lipschitz continuous with respect to $\mathsf{d}$ with constant $L$.

**Lemma 2.** *For any* $(x,y),(x',y') \in \mathbb{R}^d \times \mathbb{R}^d$, $|c(x,y) - c(x',y')| \leq L\mathsf{d}((x,y),(x',y'))$.

*Proof.* As $c(x,y) = \sum_{i=1}^d h(y_i - x_i)$ for $(x,y) \in \mathbb{R}^d \times \mathbb{R}^d$ and $h$ is Lipschitz continuous with constant $L \geq 0$, we have that, for any $x, y, x', y' \in \mathbb{R}^d$,

$$|c(x,y) - c(x',y')| \leq \sum_{i=1}^d |h(y_i - x_i) - h(y_i' - x_i')| \leq L \sum_{i=1}^d |y_i - y_i' + x_i' - x_i|$$
$$\leq L(\|y - y'\|_1 + \|x - x'\|_1).$$

$\square$

**Lemma 3.** *For any choice of* $(\mu,\nu),(\mu',\nu') \in \mathcal{P}(\mathbb{R}^d) \times \mathcal{P}(\mathbb{R}^d)$ *and* $\pi \in \Pi(\mu,\nu), \pi' \in \Pi(\mu',\nu')$, *we have that*

$$\left| \int c\,d(\pi - \pi') \right| \leq L\mathsf{W}_1(\pi, \pi'), \tag{7}$$

*where* $\mathsf{W}_1(\pi, \pi') = \mathsf{OT}_\mathsf{d}(\pi, \pi')$ *is the 1-Wasserstein distance for the metric* $\mathsf{d}$.

*Proof.* Let $(\mu, \nu), (\mu', \nu') \in \mathcal{P}(\mathbb{R}^d) \times \mathcal{P}(\mathbb{R}^d)$ be arbitrary and fix any $\pi \in \Pi(\mu, \nu)$ and $\pi' \in \Pi(\mu', \nu')$. Let $\gamma \in \Pi(\pi, \pi')$ be arbitrary and consider

$$\left| \int c(x, y) d\pi(x, y) - \int c(x', y') d\pi'(x', y') \right| = \left| \int c(x, y) - c(x', y') d\gamma(x, y, x', y') \right|,$$

$$\leq \int |c(x, y) - c(x', y')| \, d\gamma(x, y, x', y'),$$

$$\leq L \int \mathsf{d}((x, y), (x', y')) d\gamma.$$

As $\gamma$ is arbitrary, it follows that $\left| \int c d(\pi - \pi') \right| \leq \mathsf{OT}_{\mathsf{d}}(\pi, \pi')$. $\qquad \square$

The proof of Theorem 1 is based on the framework developed in [Eckstein and Nutz, 2023] and, in particular, the proof of Theorem 3.1 therein. As a byproduct of their proof technique, it is demonstrated that, for any $\pi \in \Pi(\mu, \nu)$, there exists a coupling $\pi' \in \Pi(\mu, \nu)$ which is independent of $c$ satisfying

$$\int c d\pi' - \int c d\pi \leq 2L \left( \mathsf{OT}_{|\cdot|, 0}(\mu^n, \mu) \wedge \mathsf{OT}_{|\cdot|, 0}(\nu^n, \nu) \right), \tag{8}$$

provided that the cost satisfies the condition (7) (see Definition 3.1 in [Eckstein and Nutz, 2023] for a weaker condition) for any choice of $\mu^n, \nu^n \in \mathcal{P}^n(\mathbb{R}^d)$, the set of all probability measures on $\mathbb{R}^d$ supported on at most $n$ points. We note that $\mathsf{OT}_{|\cdot|, 0}$ is simply the 1-Wasserstein distance for the distance induced by the 1-norm and recall that $|\cdot|$ has Lipschitz constant 1 due to the reverse triangle inequality (recall the notation $\mathsf{OT}_{h, \lambda}$ from Section 3). It is then shown in their proof that

$$\int c d\pi' - \int c d\pi \leq 4LC\lambda, \quad \mathsf{D}_{\mathsf{KL}}(\pi' || \mu \otimes \nu) \leq \frac{1}{\alpha} \log \left( \frac{1}{\lambda} \right), \tag{9}$$

provided that there exists $\mu^n, \nu^n \in \mathcal{P}^n(\mathbb{R}^d)$ satisfying $\left( \mathsf{OT}_{|\cdot|, 0}(\mu^n, \mu) \wedge \mathsf{OT}_{|\cdot|, 0}(\nu^n, \nu) \right) \leq Cn^{-\alpha}$ for $n = \lfloor \lambda^{-1/\alpha} \rfloor$ for some $\alpha \in (0, 1]$ and $C \geq 0$.

**Lemma 4.** *Fix $\delta > 0$ and assume that $\eta \in \mathcal{P}(\mathbb{R}^d)$ is not mutually singular with respect to the Lebesgue measure. Then, there exists a measure $\eta^n \in \mathcal{P}^n(\mathbb{R}^d)$ satisfying*

$$\mathsf{OT}_{|\cdot|, 0}(\eta^n, \eta) \leq n^{-1/d} \left( \frac{5}{\delta} \sum_{i=1}^d \mathbb{E}_\eta |X_i|^{1+\delta} + \frac{20d}{\delta} \right),$$

*for every $n \geq (20/\delta)^d$.*

*Proof.* First, note that if $\sum_{i=1}^d \mathbb{E}_\eta |X_i|^{1+\delta} = \infty$, the inequality holds vacuously. We hence assume that $\sum_{i=1}^d \mathbb{E}_\eta |X_i|^{1+\delta} < \infty$.

Fix $\delta > 0$. Then, by Lemmas 6.6 and 6.7 in [Graf and Luschgy, 2000], there exists constants $C_1, C_2, C_3 > 0$ depending on $\delta$ for which

$$\inf_{\rho \in \mathcal{P}^n(\mathbb{R}^d)} \mathsf{OT}_{|\cdot|, 0}(\rho, \eta) \leq n^{-1/d} \left( 2C_1 \sum_{i=1}^d \mathbb{E}_\eta |X_i|^{1+\delta} + 2dC_2 \right), \tag{10}$$

for every $n \geq (2C_3)^d$. Here $C_1 = \frac{5}{2\delta}$ and $C_2, C_3 > 0$ can be chosen as any constants satisfying $\frac{\Gamma(2)\Gamma(\delta n - 1)}{\Gamma(\delta n)} = \frac{1}{\delta n - 1} \leq \frac{C_2}{5n}$ for every $n \geq \frac{C_3}{5}$. Observe that $C_2 = C_3 = \frac{10}{\delta}$ satisfy these conditions.

We conclude by noting that the infimum in (10) is achieved by Theorem 4.12 in [Graf and Luschgy, 2000]. $\qquad \square$

**Lemma 5.** *Fix $\delta > 0$ and $\eta \in \mathcal{P}(\mathbb{R}^d)$. Then, there exists a measure $\eta^n \in \mathcal{P}^n(\mathbb{R}^d)$ satisfying*

$$\mathsf{OT}_{|\cdot|, 0}(\eta^n, \eta) \leq n^{-1/d} \left( \frac{5}{\delta} \sum_{i=1}^d \mathbb{E}_\eta |X_i|^{1+\delta} + \frac{20d}{\delta} \right),$$

*for every $n \geq (20/\delta)^d$.*

*Proof.* Again, if $\sum_{i=1}^{d} \mathbb{E}_\eta |X_i|^{1+\delta} = \infty$, the inequality trivially holds.

Assume that $\sum_{i=1}^{d} \mathbb{E}_\eta |X_i|^{1+\delta} < \infty$ and, for $t \geq 0$, let $g_t : \mathbb{R}^d \to \mathbb{R}$ denote the density of an isotropic mean-zero normal distribution with covariance $t^2 \mathrm{Id}$. Define the probability measure $\eta * g_t$ via

$$\eta * g_t(A) = \iint \mathbb{1}_A(x+y)g_t(y)dy d\eta(x) = \int \left( \int \mathbb{1}_A(z)g_t(z-x)dz \right) d\eta(x)$$
$$= \int \mathbb{1}_A(z) \left( \int g_t(z-x)d\eta(x) \right) dz,$$

for any Borel measurable set $A \subset \mathbb{R}^d$. From the above display, $\eta * g_t$ has density $z \in \mathbb{R}^d \mapsto \int g_t(z-x)d\eta(x)$ with respect to the Lebesgue measure.

It follows from a minor modification of Lemma 7.1.10 in [Ambrosio et al., 2005], that, for any $1 \leq p < \infty$, $\mathsf{OT}_{|\cdot|^p, 0}(\eta * g_t, \eta) \leq t^p \int \|z\|_1^p g_t(z)dz < \infty$ as normal distributions have finite absolute moments of all orders. Hence, $\lim_{t\downarrow 0} \mathsf{OT}_{|\cdot|, 0}(\eta * g_t, \eta) = 0$ and $\mathbb{E}_{\eta*g_t}|X_i|^{1+\delta} \to \mathbb{E}_\eta |X_i|^{1+\delta}$ as $t \downarrow 0$ (see Theorem 6.9 in [Villani, 2009]).

Now, let $\eta^n \in \mathcal{P}^n(\mathbb{R}^d)$ be such that

$$\mathsf{OT}_{|\cdot|, 0}(\eta^n, \eta * g_t) \leq n^{-1/d} \left( \frac{5}{\delta} \sum_{i=1}^{d} \mathbb{E}_{\eta*g_t}|X_i|^{1+\delta} + \frac{20d}{\delta} \right)$$

for every $n \geq (20/\delta)^d$ as in Lemma 4. It follows for the triangle inequality for Wasserstein distances (see Chapter 6 in Villani, 2009) that

$$\mathsf{OT}_{|\cdot|, 0}(\eta^n, \eta) \leq \mathsf{OT}_{|\cdot|, 0}(\eta^n, \eta * g_t) + \mathsf{OT}_{|\cdot|, 0}(\eta, \eta * g_t).$$

The claimed result follows by applying the upper bound from the penultimate display and taking the limit $t \downarrow 0$ on both sides of the resulting inequality. $\qquad \square$

Lemma 5 provides the worst case scaling for $\mathsf{OT}_{|\cdot|, 0}(\eta^n, \eta)$. As noted in the text, it is anticipated that $d$ can be replaced by a suitable notion of intrinsic dimension for $\eta$.

*Proof of Theorem 1.* We begin with part 1. The lower bound $0 \leq \mathsf{OT}_{h, \lambda}(\mu, \nu) - \mathsf{OT}_{h, 0}(\mu, \nu)$ is due to the fact that the Kulback-Leibler divergence is non-negative. As for the upper bound, letting $\pi \in \Pi(\mu, \nu)$ be an optimal plan for $\mathsf{OT}_{h, 0}(\mu, \nu)$, such a plan always exists due to Theorem 4.1 in [Villani, 2009]. It follows from (9) and Lemma 5 that there exists a plan $\pi' \in \Pi(\mu, \nu)$ satisfying

$$\int c d\pi' + \lambda \mathsf{D}_{\mathsf{KL}}(\pi' \| \mu \otimes \nu) - \mathsf{OT}_{h, 0}(\mu, \nu) \leq d\lambda \log\left( \frac{1}{\lambda} \right) + 4L\lambda \left( \frac{5}{\delta} \sum_{i=1}^{d} \mathbb{E}_\eta |X_i|^{1+\delta} + \frac{20d}{\delta} \right)$$

provided that $\lfloor \lambda^{-d} \rfloor \geq (20/\delta)^d$. By minimizing both sides of the above display with respect to $\pi' \in \Pi(\mu, \nu)$, Part 1 readily follows.

The proof of Part 2 will follow from the fact that the set of all couplings $\Pi(\mu, \nu)$ is tight and hence admits a limit point in the weak topology by Prokhorov's theorem (see Lemma 4.4 in Villani, 2009) and Corollary 7.20 in [Maso, 1993] once we establish that the functionals

$$\mathsf{F}_\lambda : \pi \in \mathcal{P}(\mathbb{R}^d \times \mathbb{R}^d) \mapsto \begin{cases} \int c d\pi + \lambda \mathsf{D}_{\mathsf{KL}}(\pi \| \mu \otimes \nu), & \text{if } \pi \in \Pi(\mu, \nu), \\ +\infty, & \text{otherwise,} \end{cases}$$

$$\mathsf{F} : \pi \in \mathcal{P}(\mathbb{R}^d \times \mathbb{R}^d) \mapsto \begin{cases} \int c d\pi, & \text{if } \pi \in \Pi(\mu, \nu), \\ +\infty, & \text{otherwise,} \end{cases}$$

are such that $\mathsf{F}_\lambda$ $\Gamma$-converges to $\mathsf{F}$ when treated as functionals on the separable metric space $(\mathcal{P}(\mathbb{R}^d \times \mathbb{R}^d), \mathsf{d}')$, where $\mathsf{d}'$ is the Lévy-Prokhorov metric which metrizes the weak convergence of probability distributions (cf. e.g. p.72 in [Billingsley, 2013]), we refer the reader to [Maso, 1993] as a standard reference on $\Gamma$-convergence. In light of Proposition 8.1 in [Maso, 1993] it suffices to show that:

1. for every $\pi \in \mathcal{P}(\mathbb{R}^d \times \mathbb{R}^d)$ and any sequence $\mathcal{P}(\mathbb{R}^d \times \mathbb{R}^d) \ni \pi_\lambda \to \pi$ with respect to $\mathsf{d}'$, $\mathsf{F}(\pi) \leq \liminf_{\lambda \downarrow 0} \mathsf{F}_\lambda(\pi_\lambda)$.

2. for every $\pi \in \mathcal{P}(\mathbb{R}^d \times \mathbb{R}^d)$ there exists a sequence $\mathcal{P}(\mathbb{R}^d \times \mathbb{R}^d) \ni \pi_\lambda \to \pi$ with respect to $d'$ satisfying $\mathsf{F}(\pi) = \lim_{\lambda \downarrow 0} \mathsf{F}_\lambda(\pi_\lambda)$.

We start by proving the first statement. Fix $\pi \in \mathcal{P}(\mathbb{R}^d \times \mathbb{R}^d)$ and any sequence $(\pi_\lambda)_{\lambda \downarrow 0} \subset \mathcal{P}(\mathbb{R}^d \times \mathbb{R}^d)$ converging to $\pi$ with respect to $\mathsf{W}_1$. By Lemma 4.4 in [Villani, 2009], $\Pi(\mu, \nu)$ is tight and hence precompact with respect to $d'$ by Prokhorov's theorem. It is easy to see that $\Pi(\mu, \nu)$ is closed under the weak convergence such that it is in fact compact with respect to $d'$. It follows that if $\pi \notin \Pi(\mu, \nu)$, $\pi_\lambda \notin \Pi(\mu, \nu)$ for every $\lambda$ sufficiently small, hence $\mathsf{F}_\lambda(\pi_\lambda) \to +\infty = \mathsf{F}(\pi)$ as $\lambda \downarrow 0$. Now, if $\pi \in \Pi(\mu, \nu)$, compactness of $\Pi(\mu, \nu)$ implies that $\pi_\lambda \in \Pi(\mu, \nu)$ for every $\lambda$ sufficiently small. As $\mu, \nu$ have finite first moments and $c$ is Lipschitz continuous with constant $L$, any $\gamma \in \Pi(\mu, \nu)$ satisfies

$$\int |c(x,y)| d\gamma(x,y) \leq \int |c(0,0)| + L(\|x\|_1 + \|y\|_1) d\gamma(x,y)$$
$$= |c(0,0)| + L \int \|x\|_1 d\mu(x) + L \int \|y\|_1 d\nu(y) < \infty.$$

Conclude from Lemma 5.1.7 in [Ambrosio et al., 2005] that

$$\mathsf{F}_\lambda(\pi_\lambda) = \int c d\pi_\lambda + \lambda \mathsf{D}_{\mathsf{KL}}(\pi \| \mu \otimes \nu) \geq \int c d\pi_\lambda \to \int c d\pi = \mathsf{F}(\pi),$$

proving the first condition.

As for the second condition, if $\pi \notin \Pi(\mu, \nu)$, the constant sequence $\pi_\lambda = \pi$ satisfies $\mathsf{F}_\lambda(\pi_\lambda) = \mathsf{F}_\lambda(\pi) = \mathsf{F}(\pi) = +\infty$. If $\pi \in \Pi(\mu, \nu)$, let $\mu^n, \nu^n \in \mathcal{P}^n(\mathbb{R}^d)$ be such that $\mathsf{OT}_{|\cdot|,0}(\mu^n, \mu), \mathsf{OT}_{|\cdot|,0}(\nu^n, \nu) \to 0$ as $n \to \infty$ (i.e. $\mu^n$ and $\nu^n$ converge weakly to $\mu$ and $\nu$ respectively and $\mathbb{E}_{\mu^n}[\|X\|_1] \to \mathbb{E}_\mu[\|X\|_1], \mathbb{E}_{\nu^n}[\|Y\|_1] \to \mathbb{E}_\nu[\|Y\|_1]$). For instance, the empirical versions of $\mu$ and $\nu$ constructed from independent samples satisfy this condition almost surely (see Theorem 3 in [Varadarajan, 1958] for weak convergence, convergence of the moments is due to the law of large numbers). By (8) and the surrounding discussion, there exists $\pi_\lambda \in \Pi(\mu, \nu)$ satisfying

$$\int c d\pi_\lambda - \int c d\pi \leq 4L \left( \mathsf{OT}_{|\cdot|,0}\left(\mu^{\lfloor \lambda^{-1} \rfloor}, \mu\right) \wedge \mathsf{OT}_{|\cdot|,0}\left(\nu^{\lfloor \lambda^{-1} \rfloor}, \nu\right) \right) \to 0 \quad \text{as } \lambda \downarrow 0.$$

From the proof of Theorem 3.1 in [Eckstein and Nutz, 2023], $\lambda \mathsf{D}_{\mathsf{KL}}(\pi_\lambda \| \mu \otimes \nu) \leq \lambda \log(\lfloor \lambda^{-1} \rfloor) \to 0$ as $\lambda \downarrow 0$ and

$$\mathsf{W}_1(\pi_\lambda, \pi) \leq 2 \left( \mathsf{OT}_{|\cdot|,0}\left(\mu^{\lfloor \lambda^{-1} \rfloor}, \mu\right) \wedge \mathsf{OT}_{|\cdot|,0}\left(\nu^{\lfloor \lambda^{-1} \rfloor}, \nu\right) \right) \to 0 \quad \text{as } \lambda \downarrow 0.$$

Conclude that

$$\mathsf{F}_\lambda(\pi_\lambda) = \int c d\pi_\lambda + \lambda \mathsf{D}_{\mathsf{KL}}(\pi_\lambda \| \mu \otimes \nu) \to \int c d\pi = \mathsf{F}(\pi),$$

and, as $\mathsf{W}_1(\pi_\lambda, \pi) \to 0$, $\pi_\lambda$ converges weakly to $\pi$ (see [Villani, 2009]). This concludes the proof. $\square$

### C.3 Proof of Theorem 2

We first establish some useful properties of optimal potentials for this problem, there exists a pair

**Lemma 6.** *Fix $\lambda > 0$ and sub-Gaussian distributions $\mu, \nu \in \mathcal{P}(\mathbb{R}^d)$ with a shared constant $\tau^2 > 0$. Then, for any choice of $h$ satisfying* $(\mathrm{SC_d})$, *there exists a unique pair of continuous optimal potentials $(\varphi, \psi)$ for $\mathsf{OT}_{h,\lambda}(\mu, \nu)$ for which the Schrödinger system (4) holds at all points $(x,y) \in \mathbb{R}^d \times \mathbb{R}^d$ and $\varphi(0) = 0$. Moreover, this pair of potentials satisfies the estimates*

$$|\varphi(x)| \leq C_{d,L,\tau,h(0)}(1 + \|x\|_1), \quad |\psi(y)| \leq C_{d,L,\tau,h(0)}(1 + \|y\|_1),$$
$$|D^\alpha \varphi(x)| \leq C_{d,k,\tau,\lambda,p_k,L,h(0)}(1 + \|x\|_1)^{kp_k}, \quad |D^\alpha \psi(y)| \leq C_{d,k,\tau,\lambda,p_k,L,h(0)}(1 + \|y\|_1)^{kp_k},$$

*for any multi-index $\alpha \in \mathbb{N}_0^d$ of order $|\alpha| \leq k = \lfloor d/2 \rfloor + 1$ with the constants $k, L, p_k$ from the $(\mathrm{SC_d})$ condition. The quantities in the subscripts of the constants indicate what parameters the constants depend on.*

*Proof.* Let $(\varphi_h, \psi_h)$ be an arbitrary pair of optimal potentials for $\mathsf{OT}_{h,\lambda}(\mu, \nu)$ so that

$$\int \varphi_h d\mu + \int \psi_h d\nu = \mathsf{OT}_{h,\lambda}(\mu_0, \mu_1),$$

and, setting $\mathsf{C} = \int \psi_h d\nu - \frac{1}{2}\mathsf{OT}_{h,\lambda}(\mu, \nu)$, we have that $(\varphi_h', \psi_h') = (\varphi_h + \mathsf{C}, \psi_h - \mathsf{C})$ is another pair of optimal potentials, and $\int \varphi_h' d\mu = \int \psi_h' d\nu = \frac{1}{2}\mathsf{OT}_{h,\lambda}(\mu, \nu)$.

We further consider the functions defined on $\mathbb{R}^d$ by

$$\varphi_h''(x) = -\lambda \log \left( \int e^{\frac{\psi_h'(y) - c(x,y)}{\lambda}} d\nu(y) \right), \quad \psi_h''(y) = -\lambda \log \left( \int e^{\frac{\varphi_h''(x) - c(x,y)}{\lambda}} d\mu(x) \right).$$

We will show that $(\varphi_h'', \psi_h'')$ are optimal potentials satisfying the claimed bounds.

It follows by Jensen's inequality and Lemma 2 that

$$\varphi_h''(x) \le \int c(x,y) - \psi_h'(y) d\nu(y) \le \underbrace{c(0,0)}_{=h(0)} + L \underbrace{\int \|y\|_1 d\nu(y)}_{\le \sqrt{4\tau^2}} + L\|x\|_1 - \frac{1}{2} \underbrace{\mathsf{OT}_{h,\lambda}(\mu, \nu)}_{\ge 0},$$

observing that $2 \ge \mathbb{E}_\nu[\exp(\|X\|_1^2/2\tau^2)] \ge \mathbb{E}_\nu[\|X\|_1^2/2\tau^2]$ such that $\int \|\cdot\|_1 d\nu \le \sqrt{4\tau^2}$ (again by Jensen's inequality). It follows that $\varphi_h''(x) \le C_{d,L,\tau,h(0)}(1 + \|x\|_1)$ where $C_{d,L,\tau,h(0)}$ depends on $d, L, \tau$, and the value of $h(0)$. The same bound evidently holds for $\psi_h''$ on $\mathbb{R}^d$ and for $\psi_h'$ on the support of $\nu$. Applying this bound and Lemma 2, it holds that

$$-\varphi_h''(x) \le \lambda \log \left( \int e^{\frac{C_{d,L,\tau}(1 + \|y\|_1) - h(0) + L\|x\|_1 + L\|y\|_1}{\lambda}} d\nu(y) \right) \le L\|x\|_1 + C_{d,L,\tau,h(0)}',$$

where we have used the fact that $\mathbb{E}_\nu[e^{t\|X\|_1}] \le \mathbb{E}_\nu\left[ e^{\frac{\tau^2 t^2}{2} + \frac{\|X\|_1^2}{2\tau^2}} \right] \le 2e^{\frac{\tau^2 t^2}{2}}$ for any $t \in \mathbb{R}$ as follows from Young's inequality and the sub-Gaussian assumption. The same argument implies that $\psi_h''$ satisfies an analogous bound.

To see that $(\varphi_h'', \psi_h'')$ is a pair of optimal potentials, observe that Jensen's inequality yields

$$\int (\varphi_h' - \varphi_h'') d\mu + \int (\psi_h' - \psi_h'') d\nu \le \lambda \log \int e^{\frac{\varphi_h' - \varphi_h''}{\lambda}} d\mu + \lambda \log \int e^{\frac{\psi_h' - \psi_h''}{\lambda}} d\nu$$

$$= \lambda \log \int e^{\frac{\varphi_h'(x) + \psi_h'(y) - c(x,y)}{\lambda}} d\mu \otimes \nu(x,y)$$

$$+ \lambda \log \int e^{\frac{\varphi_h''(x) + \psi_h'(y) - c(x,y)}{\lambda}} d\mu \otimes \nu(x,y) = 0$$

as follows from the fact that $(\varphi_h', \psi_h')$ satisfy (4). Conclude that $\int \varphi_h'' d\mu + \int \psi_h'' d\nu \ge \mathsf{OT}_{h,\lambda}(\mu, \nu)$ such that equality must hold (indeed $\int e^{\frac{\varphi_h''(x) + \psi_h''(y) - c(x,y)}{\lambda}} d\mu \otimes \nu(x,y) = 1$ by construction, so the left hand side of the ineqalit is the objective value in the dual form of the EOT problem) and, by strict concavity of the logarithm, $\varphi_h'' = \varphi_h'$ $\mu$-a.e. and $\psi_h'' = \psi_h'$ $\nu$-a.e. such that $(\varphi_h'', \psi_h'')$ are indeed optimal potentials for this problem.

Now, consider the potentials $(\varphi, \psi) = (\varphi_h'' - \varphi_h''(0), \psi_h'' + \varphi_h''(0))$, which satisfy (4) on $\mathbb{R}^d \times \mathbb{R}^d$ $\varphi(0) = 0$. The bounds for $\varphi_h''$ and $\psi_h''$ evidently carry over to $\varphi$ and $\psi$ so that there exists a constant $C_{d,L,\tau,h(0)}'' < \infty$ depending only on $d, L, \tau$, and $h(0)$ for which

$$|\varphi(x)| \le C_{d,L,\tau,h(0)}''(1 + \|x\|_1), \quad |\psi(y)| \le C_{d,L,\tau,h(0)}''(1 + \|y\|_1),$$

for every $(x,y) \in \mathbb{R}^d \times \mathbb{R}^d$. Now, suppose that $(\bar\varphi, \bar\psi)$ is any other pair of potentials for which (4) holds on $\mathbb{R}^d \times \mathbb{R}^d$ and $\bar\varphi(0) = 0$. As discussed in Section 3, one has that $(\varphi, \psi)$ and $(\bar\varphi, \bar\psi)$ coincide $\mu$- and $\nu$-a.e. so that, for every $x \in \mathbb{R}^d$,

$$e^{-\frac{\bar\varphi(x)}{\lambda}} = \int e^{\frac{\bar\psi(y) - c(x,y)}{\lambda}} d\nu(y) = \int e^{\frac{\psi(y) - c(x,y)}{\lambda}} d\nu(y) = e^{-\frac{\varphi(x)}{\lambda}},$$

which shows that $\bar\varphi = \varphi$, the equality of $\psi$ and $\bar\psi$ follows analogously.

We now establish the bounds on the derivatives. By the multivariate Faà di Bruno formula (cf. e.g. Constantine and Savits, 1996), for any multi-index $\alpha \in \mathbb{N}_0^d$ of order $|\alpha| \geq 1$, the derivative $-D^\alpha \varphi_h''(x)$ (assuming it exists) can be expressed as a linear combination of products of derivatives of the form

$$\prod_{j=1}^{|\alpha|} \left( \frac{D_x^{\beta_j} \int e^{\frac{\psi_h''(y)-c(x,y)}{\lambda}} d\nu(y)}{\int e^{\frac{\psi_h''(y)-c(x,y)}{\lambda}} d\nu(y)} \right)^{k_j}, \tag{11}$$

where $\beta_j \in \mathbb{N}_0^d$ and $k_j \in \mathbb{N}_0$ satisfy $\sum_{j=1}^{|\alpha|} \beta_j k_j = \alpha$. By the dominated convergence theorem, the derivative commutes with the integral sign and, applying the same formula to the derivative $D_x^{\beta_j} e^{\frac{-c(x,y)}{\lambda}}$, we see that it can be expressed as a linear combination of products of the form

$$e^{-\frac{c(x,y)}{\lambda}} \prod_{m=1}^{|\beta_j|} \left( -\frac{1}{\lambda} D_x^{\gamma_m} \sum_{i=1}^d h(y_i - x_i) \right)^{l_m}, \tag{12}$$

where $\gamma_m \in \mathbb{N}_0^d$, $l_m \in \mathbb{N}_0$ with $\sum_{m=1}^{|\beta_j|} \gamma_m l_m = \beta_j$, and we assume that all relevant derivatives exist.

As $h$ satisfies ($\mathrm{SC_d}$), it is $k$-times continuously differentiable for $k = \lfloor d/2 \rfloor + 1$ with derivatives of order $s \leq k$ satisfying $|h^{(s)}(x)| \leq C_k(1+|x|)^{p_k}$ for some $C_k < \infty$ and $p_k > 1$ which may depend on $k$. From this assumption, (12) is well defined provided that $|\beta_j| \leq k$ and we observe that $\left| D_x^{\gamma_m} \sum_{i=1}^d h(y_i - x_i) \right| \leq \sum_{i=1}^d C_k(1+|y_i - x_i|)^{p_k} \leq \sum_{i=1}^d C_k 2^{p_k-1}(1+|y_i - x_i|^{p_k})$ as $p_k > 1$, so that $\left| D_x^{\beta_j} e^{-\frac{c(x,y)}{\lambda}} \right|$ can be bounded as $e^{-\frac{c(x,y)}{\lambda}} C_{d,k,p_k,\lambda}(1+\|y-x\|_1)^{p_k|\beta_j|}$ by appealing to (12).

Returning to (11), we infer that $D^\alpha \varphi_h''$ is well-defined for any $|\alpha| \leq k$ and that

$$\left| D_x^{\beta_j} \int e^{\frac{\psi_h''(y)-c(x,y)}{\lambda}} d\nu(y) \right| = \left| \int e^{\frac{\psi_h''(y)}{\lambda}} D_x^{\beta_j} e^{-\frac{c(x,y)}{\lambda}} d\nu(y) \right|$$

$$\leq C_{d,k,p_k,\lambda} \int e^{\frac{\psi_h''(y)-c(x,y)}{\lambda}} (1+\|y-x\|_1)^{p_k|\beta_j|} d\nu(y).$$

We will split this integral into the regions $\|y\|_1 < \tau$ and $\|y\|_1 \geq \tau$ for some $\tau > 0$ which will be chosen later in the proof.

In this first region,

$$\int_{\{\|y\|_1 < \tau\}} e^{\frac{\psi_h''(y)-c(x,y)}{\lambda}} (1+\|y-x\|_1)^{p_k|\beta_j|} d\nu(y) \leq (1+\tau+\|x\|_1)^{p_k|\beta_j|} \int e^{\frac{\psi_h''(y)-c(x,y)}{\lambda}} d\nu(y),$$

by the triangle inequality so that

$$\frac{D_x^{\beta_j} \int_{\{\|y\|_1<\tau\}} e^{\frac{\psi_h''(y)-c(x,y)}{\lambda}} d\nu(y)}{\int e^{\frac{\psi_h''(y)-c(x,y)}{\lambda}} d\nu(y)} \leq (1+\tau+\|x\|_1)^{p_k|\beta_j|}$$

For the second region, we have from Lemma 2 and the first part of the proof that

$$\int_{\{\|y\|_1 \geq \tau\}} e^{\frac{\psi_h''(y)-c(x,y)}{\lambda}} (1+\|y-x\|_1)^{p_k|\beta_j|} d\nu(y)$$

$$\leq \int_{\{\|y\|_1 \geq \tau\}} e^{\frac{\psi_h''(y)-h(0)+L\|x\|_1+L\|y\|_1}{\lambda}} (1+\|y-x\|_1)^{p_k|\beta_j|} d\nu(y)$$

$$\leq \int_{\{\|y\|_1 \geq \tau\}} e^{\frac{L\sqrt{4\tau^2}+L\|x\|_1+2L\|y\|_1}{\lambda}} (1+\|y-x\|_1)^{p_k|\beta_j|} d\nu(y).$$

To bound this integral, note that

$$(1+\|y-x\|_1)^{p_k|\beta_j|} \leq (1+\|y\|_1+\|x\|_1)^{p_k|\beta_j|} \leq 2^{p_k|\beta_j|-1} \left( (1+\|x\|_1)^{p_k|\beta_j|} + \|y\|_1^{p_k|\beta_j|} \right),$$

and that

$$\int_{\{\|y\|_1\geq\tau\}} e^{\frac{L\sqrt{4\tau^2}+L\|x\|_1+2L\|y\|_1}{\lambda}} d\nu(y) = e^{\frac{L\sqrt{4\tau^2}+L\|x\|_1}{\lambda}} \int_{\{\|y\|_1\geq\tau\}} e^{\frac{2L\|y\|_1}{\lambda}} d\nu(y)$$

$$\leq e^{\frac{L\sqrt{4\tau^2}+L\|x\|_1}{\lambda}} \left(\int e^{\frac{4L\|y\|_1}{\lambda}} d\nu(y)\right)^{\frac{1}{2}} \left(\int_{\{\|y\|_1\geq\tau\}} 1 d\nu(y)\right)^{\frac{1}{2}},$$

by the Cauchy-Schwarz inequality. Similarly,

$$\int_{\{\|y\|_1\geq\tau\}} e^{\frac{L\sqrt{4\tau^2}+L\|x\|_1+2L\|y\|_1}{\lambda}} \|y\|_1^{p_k|\beta_j|} d\nu(y)$$

$$\leq e^{\frac{L\sqrt{4\tau^2}+L\|x\|_1}{\lambda}} \left(\int e^{\frac{4L\|y\|_1}{\lambda}} d\nu(y)\right)^{\frac{1}{2}} \left(\int_{\{\|y\|_1\geq\tau\}} \|y\|_1^{2p_k|\beta_j|} d\nu(y)\right)^{\frac{1}{2}}.$$

Now, $\int e^{\frac{4L\|y\|_1}{\lambda}} d\nu(y) \leq 2e^{\frac{16L^2\tau^2}{2\lambda^2}}$ due to the bound $\mathbb{E}_\nu[e^{t\|x\|_1}] \leq \mathbb{E}_\nu\left[e^{\frac{\tau^2 t^2}{2}+\frac{\|X\|_1^2}{2\tau^2}}\right] \leq 2e^{\frac{\tau^2 t^2}{2}}$ which holds for every $t \geq 0$. Further,

$$\int_{\{\|y\|_1\geq\tau\}} 1 d\nu(y) \leq e^{-\frac{\tau^2}{4\tau^2}} \int_{\{\|y\|_1\geq\tau\}} e^{\frac{\|y\|_1^2}{4\tau^2}} d\nu(y) \leq \sqrt{2}e^{-\frac{\tau^2}{4\tau^2}},$$

$$\int_{\{\|y\|_1\geq\tau\}} \|y\|_1^{2p_k|\beta_j|} d\nu(y) \leq e^{-\frac{\tau^2}{4\tau^2}} \int e^{\frac{\|y\|_1^2}{4\tau^2}} \|y\|_1^{2p_k|\beta_j|} d\nu(y) \leq \sqrt{2}e^{-\frac{\tau^2}{4\tau^2}} \left(\int \|y\|_1^{4p_k|\beta_j|} d\nu(y)\right)^{\frac{1}{2}},$$

where we have applied the Cauchy-Schwarz inequality in the second line. By sub-Gaussianity, $2 \geq \mathbb{E}_\nu[e^{\|X\|_1^2/2\tau^2}] \geq \mathbb{E}_\nu\left[\frac{\|X\|_1^{2p}}{(2\tau^2)^p p!}\right]$ for any $p \in \mathbb{N}$ so that $\sqrt{\mathbb{E}_\nu\left[\|X\|_1^{4p_k|\beta_j|}\right]} \leq \sqrt{2}(2\tau^2)^{p_k|\beta_j|}\sqrt{(2p_k|\beta_j|)!}$.

Combining all of these bounds,

$$\int_{\{\|y\|_1\geq\tau\}} e^{\frac{\psi_h''(y)-c(x,y)}{\lambda}} (1+\|y-x\|_1)^{p_k|\beta_j|} d\nu(y)$$

$$\leq 2^{p_k|\beta_j|-1}(1+\|x\|_1)^{p_k|\beta_j|} e^{\frac{L\sqrt{4\tau^2}+L\|x\|_1}{\lambda}} \sqrt{2}e^{\frac{4L^2\tau^2}{\lambda^2}} e^{-\frac{\tau^2}{8\tau^2}} \left(2^{1/4} + \sqrt{\sqrt{2}(2\tau^2)^{p_k|\beta_j|}\sqrt{(2p_k|\beta_j|)!}}\right).$$

The denominator can be bounded as $\left(\int e^{\frac{\psi_h''(y)-c(x,y)}{\lambda}} d\nu(y)\right)^{-1} = e^{\frac{\varphi_h''(x)}{\lambda}} \leq e^{\frac{h(0)+L\sqrt{4\tau^2}+L\|x\|_1}{\lambda}}$ as follows from the first part of the proof. The ratio we wish to bound thus includes the exponential term $e^{\frac{h(0)+2L\sqrt{4\tau^2}+2L\|x\|_1}{\lambda}+\frac{4L^2\tau^2}{\lambda^2}-\frac{\tau^2}{8\tau^2}}$, which we equate to 1 by setting $\tau^2 = \frac{8\tau^2}{\lambda}(h(0)+2L\sqrt{4\tau^2}+2L\|x\|_1) + \frac{32L^2\tau^4}{\lambda^2} = C_{L,\tau,\lambda,h(0)} + \frac{16\tau^2}{\lambda}L\|x\|_1$.

All in all,

$$\left|\frac{D_x^{\beta_j} \int e^{\frac{\psi_h''(y)-c(x,y)}{\lambda}} d\nu(y)}{\int e^{\frac{\psi_h''(y)-c(x,y)}{\lambda}} d\nu(y)}\right|$$

$$\leq C_{d,k,p_k,\lambda}\left(1 + \left(C_{L,\tau,\lambda,h(0)}+16\tau^2\lambda^{-1}L\|x\|_1\right)^{1/2} + \|x\|_1\right)^{p_k|\beta_j|}$$

$$+ C_{d,k,p_k,\lambda} 2^{p_k|\beta_j|-1}(1+\|x\|_1)^{p_k|\beta_j|}\sqrt{2}\left(2^{1/4}+\sqrt{\sqrt{2}(2\tau^2)^{p_k|\beta_j|}\sqrt{(2p_k|\beta_j|)!}}\right),$$

the products in (11) can thus be bounded as $C_{d,k,\tau,\lambda,p_k,L,h(0)}(1+\|x\|_1)^{p_k|\alpha|}$, for any $|\alpha| \leq k$. The same argument establishes analogous bounds for $\psi_h''$. As $(\varphi,\psi)$ coincides with $(\varphi_h'',\psi_h'')$ up to additive constants, the derivative bounds evidently transfer over. $\qquad\square$

In what follows, we always choose the unique solution for any given EOT problem described in Lemma 6.

As aforementioned, our approach to proving limit distributions is based on the functional delta method (see Appendix D for a summary of the method). As $\bar{h}$ also satisfies (SC$_\text{d}$) with Lipschitz constant $2L$, $C_k$ replaced by $2C_k$, and $\bar{h}(0) = 2h(0)$, there exists a constant $C_{d,h,\tau}$ for which all bounds from Lemma 6 hold simultaneously for some choice of potentials $(\varphi_h, \psi_h)$ for $\mathsf{OT}_{h,\lambda}(\mu, \nu)$ and $(\varphi_{\bar{h}}, \psi_{\bar{h}})$ for $\mathsf{OT}_{\bar{h},\lambda}(\mu, \nu)$. We thus instantiate the function classes

$$\mathcal{F}_{\tau,h} = \left\{ f \in \mathcal{C}^k(\mathbb{R}^d) : |D^\alpha f(x)| \le C_{d,h,\tau}(1 + \|x\|_1)^{k p_k} \text{ for } k = \lfloor d/2 \rfloor + 1, \right. $$
$$\left. \forall \alpha \in \mathbb{N}_0^d \text{ with } |\alpha| \le k, \forall x \in \mathbb{R}^d \right\},$$

and

$$\mathcal{F}_{\tau,h}^\oplus = \left\{ f \oplus g : f, g \in \mathcal{F}_{\tau,h} \right\},$$

where $f \oplus g$ is understood as the function $(x, y) \in \mathbb{R}^d \times \mathbb{R}^d \mapsto f(x) + g(y)$. We further consider the class of probability distributions

$$\mathcal{P}_\tau^\otimes = \left\{ \mu \otimes \nu : \mu, \nu \in \mathcal{P}(\mathbb{R}^d) \text{ and are } \tau^2\text{-sub-Gaussian} \right\},$$

for some $\tau > 0$. Throughout, we will treat $\mathcal{P}_\tau^\otimes$ as a subset of $\ell^\infty(\mathcal{F}_{\tau,h}^\oplus)$, the Banach space of bounded real functions on $\mathcal{F}_{\tau,h}^\oplus$ endowed with the supremum norm $\|\ell\|_{\infty, \mathcal{F}_{\tau,h}^\oplus} = \sup_{f \oplus g \in \mathcal{F}_{\tau,h}^\oplus} |\ell(f \oplus g)|$; the action of $\mu \otimes \nu \in \mathcal{P}_\tau^\otimes$ on $f \oplus g$ is given by $\mu \otimes \nu(f \oplus g) = \int f d\mu + \int g d\nu$. It is easy to see that $\mu \otimes \nu$ defines a bounded function on this function class due to the growth bounds inherent to $\mathcal{F}_{\tau,h}$.

Our approach is similar to that of Proposition 1 in [Goldfeld et al., 2024b] and, in particular, Section 6 of that work. Namely, we prove a type of directional differentiability and Lipschitz continuity for the EOT cost.

**Lemma 7.** *Fix $\lambda > 0$ and sub-Gaussian distributions $\mu, \nu, \rho, \eta \in \mathcal{P}(\mathbb{R}^d)$ with constant $\tau^2 > 0$. Then, for any choice of optimal potentials $(\varphi, \psi)$ solving $\mathsf{OT}_{h,\lambda}(\mu, \nu)$ and satisfying (4) on $\mathbb{R}^d \times \mathbb{R}^d$,*

$$\lim_{t \downarrow 0} \frac{\mathsf{OT}_{h,\lambda}(\mu + t(\rho - \mu), \nu + t(\eta - \nu)) - \mathsf{OT}_{h,\lambda}(\mu, \nu)}{t} = \int \varphi d(\rho - \mu) + \int \psi d(\eta - \nu).$$

*Proof.* For $t \in [0, 1]$, let $\mu_t := \mu + t(\rho - \mu)$ and $\nu_t := \nu + t(\eta - \nu)$ and, let $(\varphi^{(\mu_t, \nu_t)}, \psi^{(\mu_t, \nu_t)})$ denote the unique pair of optimal potentials for $\mathsf{OT}_{h,\lambda}(\mu_t, \nu_t)$ satisfying $\varphi^{(\mu_t, \nu_t)}(0) = 0$ and solving the Schrödinger system on $\mathbb{R}^d \times \mathbb{R}^d$ (see Lemma 6) and likewise for $(\varphi^{(\mu_t, \nu)}, \psi^{(\mu_t, \nu)})$ and $(\varphi^{(\mu, \nu)}, \psi^{(\mu, \nu)})$. Observe that

$$\mathsf{OT}_{h,\lambda}(\mu_t, \nu_t) - \mathsf{OT}_{h,\lambda}(\mu, \nu)$$
$$= \mathsf{OT}_{h,\lambda}(\mu_t, \nu_t) - \mathsf{OT}_{h,\lambda}(\mu_t, \nu) + \mathsf{OT}_{h,\lambda}(\mu_t, \nu) - \mathsf{OT}_{h,\lambda}(\mu, \nu)$$
$$\le \int \varphi^{(\mu_t, \nu_t)} d\mu_t + \int \psi^{(\mu_t, \nu_t)} d\nu_t - \int \varphi^{(\mu_t, \nu_t)} d\mu_t - \int \psi^{(\mu_t, \nu_t)} d\nu$$
$$+ \lambda \int e^{\frac{\varphi^{(\mu_t, \nu_t)}(x) + \psi^{(\mu_t, \nu_t)}(y) - c(x,y)}{\lambda}} d\mu_t \otimes \nu(x, y) - \lambda$$
$$+ \int \varphi^{(\mu_t, \nu)} d\mu_t + \int \psi^{(\mu_t, \nu)} d\nu - \int \varphi^{(\mu_t, \nu)} d\mu - \int \psi^{(\mu_t, \nu)} d\nu$$
$$+ \lambda \int e^{\frac{\varphi^{(\mu_t, \nu)}(x) + \psi^{(\mu_t, \nu)}(y) - c(x,y)}{\lambda}} d\mu \otimes \nu(x, y) - \lambda,$$

where we recall that the potentials satisfy the relevant Schrödinger systems (4) such that $\int e^{\frac{\varphi^{(\mu_t,\nu_t)}(x)+\psi^{(\mu_t,\nu_t)}(y)-c(x,y)}{\lambda}} d\mu_t(x) \equiv 1$ and $\int e^{\frac{\varphi^{(\mu_t,\nu)}(x)+\psi^{(\mu_t,\nu)}(y)-c(x,y)}{\lambda}} d\nu(y) \equiv 1$ on $\mathbb{R}^d$ so that

$$
\begin{aligned}
& \mathsf{OT}_{h,\lambda}(\mu_t, \nu_t) - \mathsf{OT}_{h,\lambda}(\mu, \nu) \\
& = \mathsf{OT}_{h,\lambda}(\mu_t, \nu_t) - \mathsf{OT}_{h,\lambda}(\mu_t, \nu) + \mathsf{OT}_{h,\lambda}(\mu_t, \nu) - \mathsf{OT}_{h,\lambda}(\mu, \nu) \\
& \leq \int \varphi^{(\mu_t,\nu_t)} d\mu_t + \int \psi^{(\mu_t,\nu_t)} d\nu_t - \int \varphi^{(\mu_t,\nu_t)} d\mu_t - \int \psi^{(\mu_t,\nu_t)} d\nu \\
& \quad + \int \varphi^{(\mu_t,\nu)} d\mu_t + \int \psi^{(\mu_t,\nu)} d\nu - \int \varphi^{(\mu_t,\nu)} d\mu - \int \psi^{(\mu_t,\nu)} d\nu \\
& = t \int \varphi^{(\mu_t,\nu)} d(\rho - \mu) + t \int \psi^{(\mu_t,\nu_t)} d(\eta - \nu)
\end{aligned}
\tag{13}
$$

and, analogously,

$$
\mathsf{OT}_{h,\lambda}(\mu_t, \nu_t) - \mathsf{OT}_{h,\lambda}(\mu, \nu) \geq t \int \varphi^{(\mu,\nu)} d(\rho - \mu) + t \int \psi^{(\mu_t,\nu)} d(\eta - \nu).
\tag{14}
$$

It suffices, therefore, to show pointwise convergence of all relevant potentials to $(\varphi^{(\mu,\nu)}, \psi^{(\mu,\nu)})$ in the limit $t \downarrow 0$. Here we will only show convergence of $(\varphi^{(\mu_t,\nu_t)}, \psi^{(\mu_t,\nu_t)})$, convergence of the other set of potentials follows analogously. For convenience set $(\varphi_t, \psi_t) = (\varphi^{(\mu_t,\nu_t)}, \psi^{(\mu_t,\nu_t)})$

Let $[0,1] \ni t_n \downarrow 0$ be arbitrary and fix a subsequence $t_{n'}$. By Lemma 6 we can apply the Arzelà-Ascoli theorem (cf. e.g. Theorem 4.44 in Folland, 1999) to infer that $(\varphi_t, \psi_t)$ converges to a pair of continuous functions $(\varphi, \psi)$ uniformly on compact sets and, in particular, pointwise along a further subsequence $t_{n''}$. From Lemma 6,

$$
e^{\frac{\varphi_{t_{n''}}(x) + \psi_{t_{n''}}(y) - c(x,y)}{\lambda}} \leq e^{\frac{C_{d,L,\tau}(2 + \|x\|_1 + \|y\|_1) - c(0,0) + L\|x\|_1 + L\|y\|_1}{\lambda}},
$$

where this final term is integrable with respect to any product of sub-Gaussian measures. By the dominated convergence theorem, for any $(x,y) \in \mathbb{R}^d \times \mathbb{R}^d$,

$$
e^{-\frac{\varphi_{t_{n''}}(x)}{\lambda}} = \int e^{\frac{\psi_{t_{n''}}(y) - c(x,y)}{\lambda}} d(\nu + t_{n''}(\eta - \nu))(y) \to \int e^{\frac{\psi(y) - c(x,y)}{\lambda}} d\nu(y),
$$

$$
e^{-\frac{\psi_{t_{n''}}(x)}{\lambda}} = \int e^{\frac{\varphi_{t_{n''}}(x) - c(x,y)}{\lambda}} d(\mu + t_{n''}(\rho - \mu))(x) \to \int e^{\frac{\varphi(x) - c(x,y)}{\lambda}} d\mu(x),
$$

as $t_{n''} \downarrow 0$ such that the pair $(\varphi, \psi)$ satisfies the Schrödinger system (4) pointwise and hence is a pair of optimal potentials for $\mathsf{OT}_{h,\lambda}(\mu, \nu)$ and $\varphi(0) = \lim_{t_{n''} \downarrow 0} \varphi_{t_{n''}}(0) = 0$, whence $(\varphi, \psi) = (\varphi^{(\mu,\nu)}, \psi^{(\mu,\nu)})$. Combining (13) and (14), we conclude that

$$
\lim_{t_{n''} \downarrow 0} \frac{\mathsf{OT}_{h,\lambda}(\mu_{t_{n''}}, \nu_{t_{n''}}) - \mathsf{OT}_{h,\lambda}(\mu, \nu)}{t_{n''}} = \int \varphi^{(\mu,\nu)} d(\rho - \mu) + \int \psi^{(\mu,\nu)} d(\eta - \nu),
$$

and, as the limit is independent of the choice of original subsequence it follows that

$$
\lim_{t \downarrow 0} \frac{\mathsf{OT}_{h,\lambda}(\mu_t, \nu_t) - \mathsf{OT}_{h,\lambda}(\mu, \nu)}{t} = \int \varphi^{(\mu,\nu)} d(\rho - \mu) + \int \psi^{(\mu,\nu)} d(\eta - \nu).
$$

This limit is invariant under the transformation $(\varphi^{(\mu,\nu)}, \psi^{(\mu,\nu)}) \mapsto (\varphi^{(\mu,\nu)} + C, \psi^{(\mu,\nu)} - C)$, proving the claim. □

**Lemma 8.** *Fix $\lambda > 0$ and arbitrary sub-Gaussian distributions $\rho, \eta, \rho', \eta' \in \mathcal{P}(\mathbb{R}^d)$ with a shared constant $\tau^2 > 0$. Then, we have that*

$$
|\mathsf{OT}_{h,\lambda}(\rho, \eta) - \mathsf{OT}_{h,\lambda}(\rho', \eta')| \leq \|\rho \otimes \eta - \rho' \otimes \eta'\|_{\infty, \mathcal{F}_{\tau,h}^\oplus}.
$$

*Proof.* Let $(\varphi^{(\rho,\eta)}, \psi^{(\rho,\eta)})$, $(\varphi^{(\rho',\eta')}, \psi^{(\rho',\eta')})$, and $(\varphi^{(\rho',\eta)}, \psi^{(\rho',\eta)})$ be the optimal potentials for $\mathsf{OT}_{h,\lambda}(\rho, \eta)$, $\mathsf{OT}_{h,\lambda}(\rho', \eta')$, and $\mathsf{OT}_{h,\lambda}(\rho', \eta)$ described in Lemma 6. Then, by analogy with (13) and (14)

$$
\mathsf{OT}_{h,\lambda}(\rho, \eta) - \mathsf{OT}_{h,\lambda}(\rho', \eta') \leq \int \varphi^{(\rho,\eta')} d(\rho - \rho') + \int \psi^{(\rho,\eta)} d(\eta - \eta'),
$$

$$
\mathsf{OT}_{h,\lambda}(\rho, \eta) - \mathsf{OT}_{h,\lambda}(\rho', \eta') \geq \int \varphi^{(\rho',\eta')} d(\rho - \rho') + \int \psi^{(\rho,\eta')} d(\eta - \eta'),
$$

such that

$$|\mathsf{OT}_{h,\lambda}(\rho,\eta) - \mathsf{OT}_{h,\lambda}(\rho',\eta')| \le \left| \int \varphi^{(\rho',\eta)} d(\rho'-\rho) + \int \psi^{(\rho',\eta')} d(\eta'-\eta) \right|$$

$$\bigvee \left| \int \varphi^{(\rho,\eta)} d(\rho'-\rho) + \int \psi^{(\rho',\eta)} d(\eta'-\eta) \right|.$$

As the constants from Lemma 6 (and hence the constants defining $\mathcal{F}^{\oplus}_{\tau,h}$) are independent of the choice of sub-Gaussian distributions (rather they depend only on the sub-Gaussian constant), $(\varphi^{(\rho',\eta)}, \psi^{(\rho',\eta')})$ and $(\varphi^{(\rho,\eta)}, \psi^{(\rho',\eta)})$ are elements of $\mathcal{F}^{\oplus}_{\tau,h}$ so that

$$|\mathsf{OT}_{h,\lambda}(\rho,\eta) - \mathsf{OT}_{h,\lambda}(\rho',\eta')| \le \|\rho \otimes \eta - \rho' \otimes \eta'\|_{\infty, \mathcal{F}^{\oplus}_{\tau,h}}.$$

$\square$

**Lemma 9.** *Fix $\lambda > 0$ and sub-Gaussian distributions $\mu, \nu \in \mathcal{P}(\mathbb{R}^d)$ with a shared constant $\tau^2 > 0$. Then, $\sqrt{n}(\hat{\mu}_n \otimes \hat{\nu}_n - \mu \otimes \nu) \xrightarrow{d} G_{\mu \otimes \nu}$ in $\ell^{\infty}(\mathcal{F}^{\oplus}_{\tau,h})$, where $G_{\mu \otimes \nu}$ is a tight Gaussian process in $\ell^{\infty}(\mathcal{F}^{\oplus}_{\tau,h})$ for which $G_{\mu \otimes \nu}(f \oplus g) = N(0, \mathrm{var}_{\mu}(f) + \mathrm{var}_{\nu}(g))$.*

*Proof.* From the proof of Lemma 27 in [Goldfeld et al., 2024b] (see also Lemma 8 in Nietert et al., 2021), we see that the function class $\mathcal{F}_{\tau,h}$ is $\mu$-Donsker and $\nu$-Donsker (i.e. the associated empirical processes $\sqrt{n}(\hat{\mu}_n - \mu)$ and $\sqrt{n}(\hat{\nu}_n - \nu)$ converge weakly to tight mean-zero Brownian bridge processes in $\ell^{\infty}(\mathcal{F}_{\tau,h})$ with respective covariance functions given by $(f,g) \in \mathcal{F}_{\tau,h} \times \mathcal{F}_{\tau,h} \mapsto \mathrm{cov}_{\mu}(f,g)$ and $(f,g) \in \mathcal{F}_{\tau,h} \times \mathcal{F}_{\tau,h} \mapsto \mathrm{cov}_{\nu}(f,g)$ respectively) provided that $\sum_{r=1}^{\infty} r^{d+kp_k-1} \mathbb{P}_{\mu}(\|X\| \ge r-1)^{1/2} < \infty$ for $k = \lfloor d/2 \rfloor + 1$ and likewise for $\nu$. By the Chernoff bound, $\mathbb{P}_{\mu}(\|X\| \ge r-1) \le \mathbb{E}_{\mu}\left[e^{t\|X\|}\right] e^{-t(r-1)}$ for any $t > 0$. The standard inequality $\|z\| \le \|z\|_1$ for any $z \in \mathbb{R}^d$ yields $\mathbb{E}_{\mu}\left[e^{t\|X\|}\right] \le \mathbb{E}_{\mu}\left[e^{t\|X\|_1}\right] \le 2e^{\frac{t^2\tau^2}{2}}$ (recall the proof of Lemma 6). It readily follows that the sum $\sum_{r=1}^{\infty} r^{d+kp_k-1} \mathbb{P}_{\mu}(\|X\| \ge r-1)^{1/2}$ is finite, establishing Donskerness of the class with respect to $\mu$ and hence $\nu$ by the same argument.

By independence of the samples $(X_i)_{i=1}^{n}$ and $(Y_j)_{j=1}^{n}$ we have by Example 1.4.6 in [van der Vaart and Wellner, 1996], Lemma 3.2.4 in [Dudley, 2014], and Donskerness of the class that

$$(\sqrt{n}(\hat{\mu}_n - \mu), \sqrt{n}(\hat{\nu}_n - \nu)) \xrightarrow{d} (G_{\mu}, G_{\nu}) \text{ in } \ell^{\infty}(\mathcal{F}_{\tau,h}) \times \ell^{\infty}(\mathcal{F}_{\tau,h}),$$

where $G_{\mu}$ and $G_{\nu}$ are independent tight $\mu$- and $\nu$-Brownian bridge processes. As the map $(\ell, \ell') \in \ell^{\infty}(\mathcal{F}_{\tau,h}) \times \ell^{\infty}(\mathcal{F}_{\tau,h}) \mapsto \ell \otimes \ell' \in \ell^{\infty}(\mathcal{F}^{\oplus}_{\tau,h})$ is continuous (indeed $\|\ell \otimes \ell'\|_{\infty,\mathcal{F}^{\oplus}_{\tau,h}} \le \|\ell\|_{\mathcal{F}_{\tau,h}} + \|\ell'\|_{\mathcal{F}_{\tau,h}}$), we have by the continuous mapping theorem that

$$\sqrt{n}(\hat{\mu}_n - \mu) \otimes \sqrt{n}(\hat{\nu}_n - \nu)) = \sqrt{n}\left(\hat{\mu}_n \otimes \hat{\nu}_n - \mu \otimes \nu\right) \xrightarrow{d} G_{\mu \otimes \nu} \text{ in } \ell^{\infty}(\mathcal{F}^{\oplus}_{\tau,h}),$$

where $G_{\mu \otimes \nu}(f_0 \oplus f_1) = G_{\mu}(f_0) + G_{\nu}(f_1)$ for any $f_0 \oplus f_1 \in \mathcal{F}^{\oplus}_{\tau,h}$, proving the claim. $\square$

*Proof of Theorem 2.* Throughout, we fix some $\bar{\tau} > \tau$ and observe that if $\mu, \nu$ are $\tau^2$-sub-Gaussian, then they are also $\bar{\tau}^2$-sub-Gaussian. From the proof of Proposition 1 in [Goldfeld et al., 2024b] (see also Remark 4 of the same reference), Lemmas 7 and 8 together imply that the functional $\rho \otimes \eta \in \mathcal{P}^{\otimes}_{\bar{\tau}} \mapsto \mathsf{OT}_{h,\lambda}(\rho,\eta)$ is Hadamard directionally differentiable at $\mu \otimes \nu$ tangentially to $\mathcal{P}^{\otimes}_{\bar{\tau}}$ (treated as a convex subset of $\ell^{\infty}(\mathcal{F}^{\oplus}_{\bar{\tau},h})$) with derivative $\gamma \in \mathcal{T}_{\mathcal{P}^{\otimes}_{\bar{\tau}}}(\mu \otimes \nu) \mapsto \gamma(\varphi \oplus \psi)$, where $(\varphi, \psi)$ denote any pair of optimal potentials for $\mathsf{OT}_{h,\lambda}(\mu,\nu)$ satisfying (4) on $\mathbb{R}^d \times \mathbb{R}^d$, and the derivative is defined on the tangent cone to $\mathcal{P}^{\otimes}_{\bar{\tau}}$ at $\mu \otimes \nu$ which is defined as

$$\mathcal{T}_{\mathcal{P}^{\otimes}_{\bar{\tau}}}(\mu \otimes \nu) := \left\{ \gamma \in \ell^{\infty}(\mathcal{F}^{\oplus}_{\bar{\tau},h}) : \exists \mathcal{P}^{\otimes}_{\bar{\tau}} \supset (\rho_n \otimes \eta_n)_{n \in \mathbb{N}} \to \mu \otimes \nu \text{ in } \ell^{\infty}(\mathcal{F}^{\oplus}_{\bar{\tau},h}) \text{ and } t_n \downarrow 0 \right.$$

$$\left. \text{s.t. } \gamma = \lim_{n \to \infty} \frac{\rho_n \otimes \eta_n - \mu \otimes \nu}{t_n} \right\},$$

see Appendix D for precise definitions. Note that we have identified $\mathsf{OT}_{h,\lambda}(\rho,\eta)$ with a functional on $\mathcal{P}^{\otimes}_{\bar{\tau}}$; such an identification is well-defined in light of the discussion following Proposition 1 in [Goldfeld et al., 2024b].

The same implications hold for the functional $\rho \otimes \eta \in \mathcal{P}_{\bar{\tau}}^{\otimes} \mapsto \mathsf{OT}_{\bar{h},\lambda}(\rho,\eta)$ in light of the discussion preceding Lemma 7, with corresponding derivative $\gamma \in \mathcal{T}_{\mathcal{P}_{\bar{\tau}}^{\otimes}}(\mu \otimes \nu) \mapsto \gamma(\bar{\varphi} \oplus \bar{\psi})$, where $(\bar{\varphi}, \bar{\psi})$ is any pair of optimal potentials for $\mathsf{OT}_{h,\lambda}(\mu,\nu)$ satisfying (4) on $\mathbb{R}^d \times \mathbb{R}^d$.

Note that $\varepsilon_{h,\lambda}(\mu,\nu) = f \circ \left( \mathsf{OT}_{h,\lambda}(\mu,\nu), \mathsf{OT}_{\bar{h},\lambda}(\mu,\nu) \right)$ for $f : (x,y) \in \mathbb{R} \times \mathbb{R} \mapsto {}^x/_y$ such that the chain rule (cf. e.g. Proposition 3.6 in Shapiro, 1990) guarantees that $\rho \otimes \eta \in \mathcal{P}_{\bar{\tau}}^{\otimes} \mapsto \varepsilon_{h,\lambda}(\rho,\eta)$ is Hadamard directionally differentiable at $\mu \otimes \nu$ tangentially to $\mathcal{P}_{\bar{\tau}}^{\otimes}$ with derivative

$$(\varepsilon_{h,\lambda})'_{\mu \otimes \nu} : \gamma \in \mathcal{T}_{\mathcal{P}_{\bar{\tau}}^{\otimes}}(\mu \otimes \nu) \mapsto \frac{1}{\mathsf{OT}_{\bar{h},\lambda}(\mu,\nu)}\gamma(\varphi \oplus \psi) - \frac{\mathsf{OT}_{h,\lambda}(\mu,\nu)}{\mathsf{OT}_{\bar{h},\lambda}(\mu,\nu)^2}\gamma(\bar{\varphi} \oplus \bar{\psi}), \qquad (15)$$

which is notably linear as a function of $\gamma$. It will be useful to rewrite this expression in terms of a single evaluation of $\gamma$. To this end, observe that if $f_1 \oplus f_2, g_1 \oplus g_2 \in \mathcal{F}_{\bar{\tau},h}^{\oplus}$, then so too is $(\alpha f_1 - \beta g_1) \oplus (\alpha f_2 - \beta g_2)$ for any $\alpha, \beta \in \mathbb{R}$ with $|\alpha| + |\beta| \leq 1$. Moreover, setting $M = \frac{1}{\mathsf{OT}_{\bar{h},\lambda}(\mu,\nu)} \vee \frac{\mathsf{OT}_{h,\lambda}(\mu,\nu)}{\mathsf{OT}_{\bar{h},\lambda}(\mu,\nu)^2}$ (which is assumed to be nonzero by construction), we have that $0 \leq \frac{1}{2M}\frac{1}{\mathsf{OT}_{\bar{h},\lambda}(\mu,\nu)} \vee \frac{1}{2M}\frac{\mathsf{OT}_{h,\lambda}(\mu,\nu)}{\mathsf{OT}_{\bar{h},\lambda}(\mu,\nu)^2} \leq \frac{1}{2}$ so that we can write

$$\frac{1}{\mathsf{OT}_{\bar{h},\lambda}(\mu,\nu)}(\varphi \oplus \psi) - \frac{\mathsf{OT}_{h,\lambda}(\mu,\nu)}{\mathsf{OT}_{\bar{h},\lambda}(\mu,\nu)^2}(\bar{\varphi} \oplus \bar{\psi})$$

$$= 2M\left( \frac{1}{2M}\frac{1}{\mathsf{OT}_{\bar{h},\lambda}(\mu,\nu)}\varphi \oplus \frac{1}{2M}\frac{1}{\mathsf{OT}_{\bar{h},\lambda}(\mu,\nu)}\psi \right)$$

$$- 2M\left( \frac{1}{2M}\frac{\mathsf{OT}_{h,\lambda}(\mu,\nu)}{\mathsf{OT}_{\bar{h},\lambda}(\mu,\nu)^2}\bar{\varphi} \oplus \frac{1}{2M}\frac{\mathsf{OT}_{h,\lambda}(\mu,\nu)}{\mathsf{OT}_{\bar{h},\lambda}(\mu,\nu)^2}\bar{\psi} \right)$$

$$= 2M\left( \frac{1}{2M}\frac{1}{\mathsf{OT}_{\bar{h},\lambda}(\mu,\nu)}\varphi - \frac{1}{2M}\frac{\mathsf{OT}_{h,\lambda}(\mu,\nu)}{\mathsf{OT}_{\bar{h},\lambda}(\mu,\nu)^2}\bar{\varphi} \right.$$

$$\left. \oplus \frac{1}{2M}\frac{1}{\mathsf{OT}_{\bar{h},\lambda}(\mu,\nu)}\psi - \frac{1}{2M}\frac{\mathsf{OT}_{h,\lambda}(\mu,\nu)}{\mathsf{OT}_{\bar{h},\lambda}(\mu,\nu)^2}\bar{\psi} \right),$$

where the final term in brackets is an element of $\mathcal{F}_{\bar{\tau},h}^{\oplus}$. Further, for any $\gamma \in \mathcal{T}_{\mathcal{P}_{\bar{\tau}}^{\otimes}}(\mu \otimes \nu)$, there exists a sequence $(\rho_n \otimes \eta_n)_{n \in \mathbb{N}} \subset \mathcal{P}_{\bar{\tau}}^{\otimes}$ which converges to $\mu \otimes \nu$ in $\ell^{\infty}(\mathcal{F}_{\bar{\tau},h}^{\oplus})$ and a sequence $t_n \downarrow 0$ for which $\gamma = \lim_{n \to \infty} \frac{\rho_n \otimes \eta_n - \mu \otimes \nu}{t_n}$. Thus, if $f_0 \oplus f_1, g_0 \oplus g_1 \in \mathcal{F}_{\bar{\tau},h}^{\oplus}$ are such that $f_0 + g_0 \oplus f_1 + g_1 \in \mathcal{F}_{\bar{\tau},h}^{\oplus}$, we have that

$$\gamma(f_0 \oplus f_1) + \gamma(g_0 \oplus g_1)$$

$$= \lim_{n \to \infty} t_n^{-1}\left( \int f_0 d\rho_n + \int f_1 d\eta_n - \int f_0 d\rho - \int f_1 d\eta \right)$$

$$+ \lim_{n \to \infty} t_n^{-1}\left( \int g_0 d\rho_n + \int g_1 d\eta_n - \int g_0 d\rho - \int g_1 d\eta \right)$$

$$= \lim_{n \to \infty} t_n^{-1}\left( \int f_0 + g_0 d\rho_n + \int f_1 + g_1 d\eta_n - \int f_0 + g_0 d\rho - \int f_1 + g_1 d\eta \right)$$

$$= \gamma(f_0 + g_0 \oplus f_1 + g_1).$$

Likewise, if $\alpha f_0 \oplus \alpha f_1 \in \mathcal{F}_{\bar{\tau},h}^{\oplus}$ for some $\alpha \in \mathbb{R}$, and $f_0 \oplus f_1 \in \mathcal{F}_{\bar{\tau},h}^{\oplus}$, then

$$\gamma(\alpha f_0 \oplus \alpha f_1) = \alpha \lim_{n \to \infty} t_n^{-1}\left( \int f_0 d\rho_n + \int f_1 d\eta_n - \int f_0 d\rho - \int f_1 d\eta \right) = \alpha\gamma(f_0 \oplus f_1).$$

With this, (15) can be written as

$$(\varepsilon_{h,\lambda})'_{\mu \otimes \nu} : \gamma \in \mathcal{T}_{\mathcal{P}_{\bar{\tau}}^{\otimes}}(\mu \otimes \nu) \mapsto$$

$$2M\gamma\left( \frac{1}{2M\mathsf{OT}_{\bar{h},\lambda}(\mu,\nu)}\varphi - \frac{\mathsf{OT}_{h,\lambda}(\mu,\nu)}{2M\mathsf{OT}_{\bar{h},\lambda}(\mu,\nu)^2}\bar{\varphi} \oplus \frac{1}{2M\mathsf{OT}_{\bar{h},\lambda}(\mu,\nu)}\psi - \frac{\mathsf{OT}_{h,\lambda}(\mu,\nu)}{2M\mathsf{OT}_{\bar{h},\lambda}(\mu,\nu)^2}\bar{\psi} \right).$$

$$(16)$$

Given the above differentiability result and Lemma 9, part 1 of Theorem 2 will follow from the functional delta method (see Lemma 10 in Appendix D) upon showing that $\hat{\mu}_n \otimes \hat{\nu}_n \in \mathcal{P}_{\bar{\tau}}^{\otimes}$ with probability approaching one and noting that $G_{\mu \otimes \nu} \in \mathcal{T}_{\mathcal{P}_{\bar{\tau}}^{\otimes}}(\mu \otimes \nu)$ with probability one as follows from the portmanteau theorem. To this end, note that, by the law of large numbers,

$$\mathbb{E}_{\hat{\mu}_n}\left[\exp(\|X\|_1^2/2\bar{\tau}^2)\right] = \frac{1}{n}\sum_{i=1}^n \exp(\|X_i\|_1^2/2\bar{\tau}^2) \to \mathbb{E}_\mu\left[\exp(\|X\|_1^2/2\bar{\tau}^2)\right] \leq 2^{\frac{\tau^2}{\bar{\tau}^2}} < 2.$$

almost surely such that $\hat{\mu}_n$ is $\bar{\tau}^2$-sub-Gaussian with probability approaching one. The same deliberations imply that $\hat{\nu}_n$ share the same property.

By applying the delta method, we obtain that

$$\sqrt{n}(\varepsilon_{h,\lambda}(\hat{\mu}_n, \hat{\nu}_n) - \varepsilon_{h,\lambda}(\mu, \nu)) \xrightarrow{d} (\varepsilon_{h,\lambda})'_{\mu \otimes \nu}(G_{\mu \otimes \nu}),$$

and using the explicit expression for the derivative from (16), we see that $(\varepsilon_{h,\lambda})'_{\mu \otimes \nu}(G_{\mu \otimes \nu})$ is equal in distribution to $2MN(0, v^2 + w^2)$, where $v^2 = \text{var}_\mu\left(\frac{1}{2\text{MOT}_{\bar{h},\lambda}(\mu,\nu)}\varphi - \frac{\text{OT}_{h,\lambda}(\mu,\nu)}{2\text{MOT}_{\bar{h},\lambda}(\mu,\nu)^2}\bar{\varphi}\right)$ and $w^2 = \text{var}_\nu\left(\frac{1}{2\text{MOT}_{\bar{h},\lambda}(\mu,\nu)}\psi - \frac{\text{OT}_{h,\lambda}(\mu,\nu)}{2\text{MOT}_{\bar{h},\lambda}(\mu,\nu)^2}\bar{\psi}\right)$; the $2M$ terms in the variance and multiplying the normal distribution evidently cancel out to give the desired formula for the limiting variance.

As for the bootstrap consistency result, since (15) is linear, it follows from Corollary 1 in [Goldfeld et al., 2024b] that $(\rho, \eta) \mapsto \varepsilon_{h,\lambda}(\rho, \eta)$ is Hadamard directionally differentiable at $\mu \otimes \nu$ tangentially to $\text{spt}(G_{\mu \otimes \nu})$. As in the proof of Lemma 9, the class $\mathcal{F}_{\bar{\tau}}$ is $\mu$- and $\nu$-Donsker such that the bootstrapped empirical processes $\sqrt{n}(\hat{\mu}_n^B - \hat{\mu}_n)$ and $\sqrt{n}(\hat{\nu}_n^B - \hat{\nu}_n)$ are asymptotically measurable and converge conditionally in distribution to the $\mu$- and $\nu$-Brownian bridge processes $G_\mu$ and $G_\nu$ respectively (see Chapter 3.6 in van der Vaart and Wellner, 1996). By Lemma 1.4.4 and Example 1.4.6 in [van der Vaart and Wellner, 1996], $(\sqrt{n}(\hat{\mu}_n^B - \hat{\mu}_n), \sqrt{n}(\hat{\nu}_n^B - \hat{\nu}_n))$ is asymptotically measurable and converges conditionally in distribution to $(G_\mu, G_\nu)$ as elements of $\ell^\infty(\mathcal{F}_{\bar{\tau}}) \times \ell^\infty(\mathcal{F}_{\bar{\tau}})$. As the map $(\ell, \ell') \in \ell^\infty(\mathcal{F}_{\bar{\tau}}) \times \ell^\infty(\mathcal{F}_{\bar{\tau}}) \mapsto \ell \otimes \ell' \in \ell^\infty(\mathcal{F}_{\bar{\tau},h}^{\oplus})$ is continuous, $(\sqrt{n}(\hat{\mu}_n^B - \hat{\mu}_n) \otimes \sqrt{n}(\hat{\nu}_n^B - \hat{\nu}_n))$ is asymptotically measurable and converges conditionally in distribution to $G_{\mu \otimes \nu}$ as elements of $\ell^\infty(\mathcal{F}_{\bar{\tau},h}^{\oplus})$ where $G_{\mu \otimes \nu}(f_0 \oplus f_1) = G_\mu(f_0) + G_\nu(f_1)$ for any $f_0 \oplus f_1 \in \mathcal{F}_{\bar{\tau},h}^{\oplus}$.

Bootstrap consistency then follows from Theorem 23.9 in [Van der Vaart, 2000] by applying the logic from the first half of the proof. $\qquad\square$

## D The Functional Delta Method

Our strategy for deriving limit distributions and consistency of the bootstrap is based upon the functional delta method, which generalizes the standard delta method for functions of simple random variables. This section provides a brief introduction to the functional delta method following the exposition of [Römisch, 2006]. Throughout, convergence in distribution is understood in the sense of Hoffmann-Jørgensen when necessary (cf. e.g. Chapter 1 in [van der Vaart and Wellner, 1996]).

Much like the delta method which, identifies the distributional limit of $\sqrt{n}(g(X_n) - g(\mu))$ as $N(0, \sigma^2(g'(\mu))^2)$ $n \to \infty$ provided that $\sqrt{n}(X_n - \mu) \xrightarrow{d} N(0, \sigma^2)$ and that $g : \mathbb{R} \to \mathbb{R}$ is differentiable at $\mu$ (see Proposition 8.14 in [Keener, 2010]), the functional delta method establishes the limit distribution of a functional $f : \Theta \subset D \to \mathbb{R}$, where $D$ is a normed vector space. In this setting, the surrogate for the derivative in the standard delta method is the Hadamard directional derivative.

**Definition 4** (Hadamard directional derivative [Römisch, 2006, Shapiro, 1990])**.** *Let $D$ be a normed vector space and fix a non-empty set $\Theta \subset D$. The tangent cone to $\Theta$ at $\theta \in \Theta$ is given by*

$$\mathcal{T}_\Theta(\theta) := \left\{h \in D : h = \lim_{n \to \infty} \frac{\theta_n - \theta}{t_n}, \text{ for some } \theta_n \in \Theta, \theta_n \to \theta, t_n \downarrow 0\right\}.$$

*A functional $f : \Theta \to \mathbb{R}$ is Hadamard directionally differentiable at $\theta \in \Theta$ tangentially to $\Theta$ if there exists a map $f'_\theta : \mathcal{T}_\Theta(\theta) \to \mathbb{R}$ satisfying*

$$\lim_{n \to \infty} \frac{f(\theta + t_n h_n) - f(\theta)}{t_n} = f'_\theta(h), \tag{17}$$

*for any $h \in \mathcal{T}_\Theta(\theta)$, $t_n \downarrow 0$, and $h_n \to h$ in $D$ with $\theta + t_n h_n \in \Theta$.*

This notion of differentiability is compatible with distributional convergence of random elements of $D$ in the sense that the following generalization of the delta method holds.

**Lemma 10** (Functional delta method [Römisch, 2006, Shapiro, 1991]). *Fix a probability space $(\Omega, \Sigma, \mathbb{P})$ and let $D$ be a normed vector space and $f : \Theta \subset D \to \mathbb{R}$ be Hadamard directionally differentiable at $\theta \in \Theta$ tangentially to $\mathcal{T}_\Theta(\theta)$ with derivative $f'_\theta : \mathcal{T}_\Theta(\theta) \to \mathbb{R}$. Let $T_n : \Omega \to \Theta$ be maps such that $r_n(T_n - \theta) \overset{d}{\to} T$ for some norming sequence $r_n \to \infty$ and a measurable map $T : \Omega \to \mathcal{T}_\Theta(\theta) \subset D$. Then $r_n\big(f(T_n) - f(\theta)\big) \overset{d}{\to} f'_\theta(T)$ and, if $\Theta$ is convex, $r_n\big(f(T_n) - f(\theta)\big) - f'_\theta\big(r_n(T_n - \theta)\big) \to 0$ in outer probability.*

Whilst Lemma 10 is sufficient to derive limit distributions, bootstrap consistency typically requires the following notion of full Hadamard differentiability (see e.g. Theorem 23.9 in [Van der Vaart, 2000] or Theorem 3.9.11 in van der Vaart and Wellner, 1996). A functional $f : D \to \mathbb{R}$ is said to be Hadamard differentiable at $\theta$ tangentially to a vector subspace $D_0 \subset D$ if there exists a continuous linear functional $f'_\theta : D_0 \mapsto \mathbb{R}$ satisfying (17) for any $h \in \mathcal{T}_\Theta(\theta)$, $t_n \neq 0$, $t_n \to 0$, and $h_n \to h$ in $\mathfrak{D}$ with $\theta + t_n h_n \in \Theta$. The following lemma enables a connection between full and directional Hadamard differentiability.

**Lemma 11** (Lemma 2 in [Goldfeld et al., 2024b]). *If $f : \Theta \subset D \to \mathbb{R}$ is Hadamard directionally differentiable at $\theta \in \Theta$ tangentially to $\mathcal{T}_\Theta(\theta)$ and $f'_\theta$ is linear on a subspace $D_0 \subset \mathcal{T}_\Theta(\theta)$, then $f$ is Hadamard differentiable at $\theta$ tangentially to $\mathfrak{D}_0$.*

