# OpenReview forum: "Multivariate Stochastic Dominance via Optimal Transport and Applications to Models Benchmarking"
_NeurIPS.cc/2024/Conference — NeurIPS 2024 poster_

### Official Review · Reviewer_a9kp · 2024-07-11

**Soundness:** 3
**Presentation:** 3
**Contribution:** 3
**Rating:** 6
**Confidence:** 3

**Summary:**

This paper considers the task of estimating the degree of stochastic dominance between two multivariate distributions. In the univariate setting, stochastic dominance is a useful tool for tasks such as benchmarking LLMs, where a practitioner may have estimates of some quality metric for responses. An efficient way to approximately estimate multivariate stochastic dominance could thus be helpful in setting where models may be evaluated on many metrics, rather than just a single value. The authors introduce a natural definition of near-stochastic dominance based on an existing definition in the univariate case and a connection to optimal transport, and show that while this notion may not allow for efficient estimation, adding an entropic regularization term allows for better guarantees. Through experiments, the authors demonstrate that their proposed metric converges to the true unregularized value in synthetic experiments, and for an LLM dataset, their approach outputs a ranking of models far more correlated with a "ground-truth" ranking.

**Strengths:**

- The paper is clearly written, and does a good job of explaining how their results rest on prior results in univariate stochastic domination and regularization methods.

- The goal of obtaining good methods for estimating multivariate stochastic domination is very natural, and seems very relevant to many areas of practice, especially evaluating LLMs on multiple axes, as the authors mention.

**Weaknesses:**

- The experiments section is a bit confusing, in particular the explanations of the various approaches tested in the LLM experiment. The reasoning behind the choice of the ground-truth ranking is also a bit unclear to me.

**Questions:**

- Does your method also improve estimates of the degree of stochastic dominance (e.g. the first-ranked model is much better than the second-ranked model, but the second- and third-ranked models are comparable)? Is this something that can be demonstrated empirically?

- If GPT-generated prompt comparisons are viewed as good enough to use as a ground-truth ranking, why not use this as the gold-standard benchmarking method? If not, is there a better baseline to compare the multivariate FSD approach to?

- Moreover, is there a reason why we should expect a "human" ranking to match the true stochastic order induced by some set of metrics? It seems that in some cases we might actually want the order induced by a set of metrics to look somewhat different than what the average human might output as a ranking.

**Limitations:**

I think the authors adequately address the limitations of their work.

---

> ### Author Rebuttal · Authors · 2024-08-06
>
> We thank the reviewer for their comments regarding the experiments. We will work to clarify the experimental setup and address the comments regarding comparison to ChatGPT. We believe that both of these are important to further improve the quality of our submission.
> ___
> **The experiments section is a bit confusing, in particular the explanations of the various approaches tested in the LLM experiment. The reasoning behind the choice of the ground-truth ranking is also a bit unclear to me.
> If GPT-generated prompt comparisons are viewed as good enough to use as a ground-truth ranking, why not use this as the gold-standard benchmarking method? If not, is there a better baseline to compare the multivariate FSD approach to?**
> ___
>
> We thank the reviewer for pointing out that the LLM benchmarking experiment should be more clearly explained to avoid ambiguities. We will amend this in the text. The experiment is conducted as follows:
>
> 1. We use the dataset from [1], which compiles a training set of 100K prompts along with the resulting response from 12 different LLMs.
>
> 2. For each prompt, the responses from the 12 different LLMs are evaluated according to 9 automatic metrics (e.g. BLEU, ROUGE, BERTScore, BARTScore, etc) and are stored as 12 vectors in $\mathbb R^9$ $(x^{(i)})_{i=1}^{12}$, where $x^{(1)}$ is the vector containing the outputs of the 9 metrics for the 1st LLM and so on. $x^{(1)}$ is then viewed as a sample from the distribution of all possible responses from the 1st LLM evaluated according to these 9 metrics.
>
> 3. This process is repeated for each of the 100K prompts so that one can construct the empirical measure $\hat \mu_N^{(1)}=\frac{1}{N}\sum_{i=1}^N\delta_{x^{(1)}_i}$ for $N = 100K$ (here $\delta_x$ is the Dirac measure at the point $x\in\mathbb R^9$).
>
> 4. The empirical measures $\hat \mu_n^{(i)}$ and $\hat \mu_n^{(j)}$ for some $n\leq 100K$ and $i\neq j$ are then compared by computing the normalized index $\varepsilon_{h,\lambda}(\hat \mu_n^{(i)},\hat \mu_n^{(j)})=\varepsilon_{ij}^{(h,\lambda)}$.
>
> 5. Given the pairwise ratios $\varepsilon_{ij}^{(h,\lambda)}$, we rank the 12 LLMS according to the relative testing procedure described in Section 4.2. The resampling procedure (bootstrap) is used to construct the confidence intervals.
>
> 6. The results thus obtained are compared (via Kendall Tau Similarity) to a univariate ranking based on ChatGPT scoring i.e. ChatGPT is presented by the instruction and the response of each LLM as described in point 1 and produces a score that judges the quality of the response in terms of following the given instruction; the different models are then ranked according to their ChatGPT score (a univariate ranking) and compared to the multivariate ranking described above.
>
>
> In the experiments, ChatGPT is being used for the purposes of evaluation only, motivated by its use as a human proxy in the context of evaluating the quality of generated natural language [1-4] . We show that the ranking resulting from multivariate stochastic dominance on automatic metrics correlates well with the ChatGPT ranking which, in turn, correlates well with human evaluation. The advantage of our multivariate stochastic dominance over ChatGPT or human evaluation is threefold. First, as discussed previously, it requires significantly less computational overhead. Second, there are no upfront monetary costs associated with evaluating the ratio statistic, this can be an important consideration for large-scale comparison tasks. Third, our approach can be run locally thereby eliminating the privacy concerns of exposing sensitive data on APIs running LLM as a judge, such as ChatGPT.
>
> ___
> **Moreover, is there a reason why we should expect a "human" ranking to match the true stochastic order induced by some set of metrics? It seems that in some cases we might actually want the order induced by a set of metrics to look somewhat different than what the average human might output as a ranking.**
> ___
> This is a great and subtle point. The notion of an “ideal” ranking according to a given set of metrics depends largely on the preferences of the practitioner. One limitation with the approach discussed in the text is that all dimensions are treated equally, as the same function $h$ is used to compare each dimension. This may limit the utility of this methodology in applications where violations of the order in one particular dimension should be severely penalized (e.g. unsafe responses from LLMs). In such a case, one may modify the proposed framework to use a cost function of the form $c(x,y)=\sum_{i=1}^d h_i(y_i-x_i)$ (i.e. we prescribed a different cost to violations in each dimension). Provided that each of the $h_i$ are nonnegative functions satisfying the smoothness condition, all of the theoretical results derived in the paper still go through. As such, a practitioner that wishes to formulate a domain-specific notion of almost stochastic dominance can adopt a data-driven approach to learn a new stochastic order tailored to the user’s preferences by letting the $h_i$ be parametrized costs (e.g. the logistic function) and optimizing over the parameters (in the previous example, the gain).

---

> ### Author Response · Authors · 2024-08-06
> **Rebuttal continued (1/1)**
>
> ___
> **Does your method also improve estimates of the degree of stochastic dominance (e.g. the first-ranked model is much better than the second-ranked model, but the second- and third-ranked models are comparable)? Is this something that can be demonstrated empirically?**
> ___
> This is an interesting question. Currently, the paper only addresses the question of establishing a ranking of different models according to the notion of multivariate stochastic dominance by computing the value of the ratio between each pair of models (values closer to $0$ indicate stronger dominance). Therefore, the  magnitude of the ratio is indicative of how strong or weak the dominance is with 0 indicating perfect dominance and 1 indicating the opposite; these values thus give a relative sense of dominance strength. We have included a table of the one-versus-all violation ratios computed in the application to LLM benchmarking in the rebuttal pdf which indicates how these models compare in terms of the average pairwise ratio value.
>
>
>  However, we may consider going beyond computing just the numerical value of the index to enable a more refined comparison between two models.
> Indeed, when computing the numerator via Sinkhorn’s algorithm, we obtain not only the value of the entropic optimal transport, but also an optimal plan $\pi$ for that problem which, in the discrete case considered here, characterizes how much mass should be sent from $x^{(i)}$ to $y^{(j)}$ for every pair of support points for $\mu$ and $\nu$ respectively to achieve the optimal transport. With this, we may characterize the points at which $\mu$ fails to dominate $\nu$ (i.e. points where $\pi(\{x^{(i)},y^{(j)}\})>0$ but $x^{(i)}$ does not dominate $y^{(j)}$ componentwise). This allows us to define regions where the stochastic domination of model A over model B is violated. Given such a pair $x^{(i)},y^{(j)}$, we can discern which metrics model B outperformed model A in as well as the magnitude in terms of difference of metric values by which the responses differed. This can enable us to identify which prompts lead to unsatisfactory responses from model A which can serve as a starting point for improving the overall quality of its responses.
>
>
> [1] Hada, R., Gumma, V., de Wynter, A., Diddee, H., Ahmed, M., Choudhury, M., ... & Sitaram, S. (2023). Are large language model-based evaluators the solution to scaling up multilingual evaluation?. arXiv preprint arXiv:2309.07462.
>
> [2] Jiang, D., Ren, X., & Lin, B. Y. (2023). Llm-blender: Ensembling large language models with pairwise ranking and generative fusion. arXiv preprint arXiv:2306.02561.
>
> [3] Liu, Y., Iter, D., Xu, Y., Wang, S., Xu, R., & Zhu, C. (2023). G-eval: Nlg evaluation using gpt-4 with better human alignment. arXiv preprint arXiv:2303.16634.
>
> [4] Zheng, L., Chiang, W.L., Sheng, Y., Zhuang, S., Wu, Z., Zhuang, Y., Lin, Z., Li, Z., Li, D., Xing, E. and Zhang, H. (2024). Judging llm-as-a-judge with mt-bench and chatbot arena. Advances in Neural Information Processing Systems, 36.

---

> > ### Comment · Reviewer_a9kp · 2024-08-10
> >
> > Thanks for your detailed response to my review. I will maintain my score as I think the main results are timely and of use to the LLM community, but the paper could probably benefit from a more thorough experiments section and improved clarity of writing throughout.

---

### Official Review · Reviewer_RTCD · 2024-07-11

**Soundness:** 3
**Presentation:** 3
**Contribution:** 3
**Rating:** 6
**Confidence:** 3

**Summary:**

Insipred by the uni-dimensional case of First order Stochastical Dominance (FSD), the authors indtroduce multidimensional FSD which does away making approximations using aggregations and reductions of multi dimensional metrics and reduce the orderings to unidimensional case. This is done vi Optimal Transport framework but this comes with a caveat since, using empirical Optimal Transport framework in multi dimensional case suffers from the curse of dimensionality. This is dealt with using  approximations in form of entropic regularization using ideas from Cuturi 2013, to get Entropic Optimal Transport framework.
Theorem 1 in the paper shows that the approximation error has a computable upper bound. The framework is tested on a simulated data and LLM metric evaluation case study.

**Strengths:**

1. It is great to see a method which can help LLM research community regarding proper evaluation and benchmarking. There is clearly a need for such metrics and methods, since most LLM's responses are highly stochastic, even when using same prompt and same parameters.
2. It is now high time for LLM research community to turn to rigorous statistical analysis of their research methodology and this paper contributes to that so it is both timely and useful.
3. The authors have applied the framework to two different domains: financial domain with portfolio allocation study and LLM evaluation.
4. Clear comparison between OT and EOT frameworks, absoulute and relative tests.
5. Figures are clear and support the claims made.

**Weaknesses:**

- Unclear writing in many parts of the paper which makes such a difficult technical paper harder to read and understand. I strongly advise the authors to take couple of polishing passes.
- Line 287-290 unclear writing
- Line 314-319 unclear
- predicts linearly p, indicating that it is captures well
- Figure 1 caption and title don't match.
- Maybe in experiments use multiple LLMs and not just Chat GPT

**Questions:**

Some typos and writing mistakes
1. Some references are repeated, - Del Barrio 2018
2.  How to tune the hyperparameters: $\beta$ and $\lambda$ in Sec 5.2 ?
3.  It will be helpful to clearly state that for Kendall tau statistic, lower values are better Fig 2.
4. How reasonable it is to take consensus with ChatGPT as the ground truth ?

**Limitations:**

I am not an expert in theory and this paper is heavy on OT theory, and the readibility of the paper is low in many important places. It would have been great if the authors would have listed clearly what limitations do these methods bring, one can ofcourse be related to computation when using multidimenional statistics over aggregation to uni-dimensional case, at what dimension could the cost become too prohibitive ?

---

> ### Author Rebuttal · Authors · 2024-08-06
>
> We thank the reviewer for their support of this work. We take your feedback regarding unclear writing very seriously and will further polish the paper to minimize any ambiguities. We have corrected all of the typos you have identified and reworked the sections that were identified as unclear.
>
> ___
> **How to tune the hyperparameters: $\beta$ and $\lambda$ in Sec 5.2?**
> ___
>
> This is indeed an important point when implementing this methodology, we will expand on the discussion in the main text according to the following points.
>
> The parameter $\lambda$ corresponds to the regularization strength in the entropic optimal transport distance, which is employed as a statistically efficient and computationally tractable proxy for the standard optimal transport distance. Given Theorem 1, it is desirable to set $\lambda$ as small as possible in order to approximate the true optimal transport distance well, for example $\lambda = 0.1$ as used in the numerical experiments. We remark that for practical purposes, $\lambda$ cannot be chosen arbitrarily small due to possible underflow in the matrix $e^{-C/\lambda}$, where $C$ is the matrix of pairwise costs, used when computing the entropic distance using Sinkhorn’s algorithm. Notably, Sinkhorn’s algorithm is a method tailored to positive matrices and cannot cope with zero entries. As such, a practitioner can start with a small value for $\lambda$ and increase it gradually if numerical instability is encountered.
>
> As for the choice of $\beta$, it is mentioned in Example 1 and demonstrated empirically in Section 5.1 that the larger the gain, $\beta$, of the logistic function is, the closer the logistic function is to being compatible with standard multivariate FSD as described in Definition 1. For the matrix $e^{-C/\lambda}$ from the previous paragraph to be well-conditioned, there is a tradeoff between $\beta$ and $\lambda$. In the experiments, these hyperparameters were set by first fixing the entropic parameter to $\lambda = 0.1$, then $\beta$ was set as large as possible subject to a computational budget constraint. Indeed, as described in the final point of this response, the number of iterations required for Sinkhorn’s algorithm to converge scales as $||C||_{\infty}/\lambda$.
> ___
> **How reasonable it is to take consensus with ChatGPT as the ground truth ?**
> ___
>
> In the experiments, the usage of ChatGPT for the purposes of evaluation is motivated by its use as a human proxy in the context of evaluating the quality of generated natural language  [2-5]. Such evaluations are common in the literature and we do not anticipate significant differences by using another equivalently sized LLM such as Claude, LLama2 70b etc. for the purpose of scoring.
>
> We underscore that the advantage of our multivariate stochastic dominance over the ChatGPT-based scoring method or human evaluation is three-fold. First, it requires significantly less computational overhead as discussed in the following point. Second, there are no upfront monetary costs associated with evaluating the ratio statistic, this can be an important consideration for large-scale comparison tasks. Third, our approach can be run locally thereby eliminating the privacy concerns of exposing sensitive data on APIs running LLM as a judge, such as ChatGPT.

---

> ### Author Response · Authors · 2024-08-06
> **Rebuttal continued (1/1)**
>
> ___
> **It would have been great if the authors would have listed clearly what limitations do these methods bring, one can ofcourse be related to computation when using multidimenional statistics over aggregation to uni-dimensional case, at what dimension could the cost become too prohibitive ?**
> ___
>
> This is another good point which deserves additional discussion in the appendix. Starting from the question of computational complexity, the dimension of the statistic is not a big concern given that the entropic distances defining the ratio are computed using Sinkhorn’s algorithm.
>
> Precisely, to solve an entropic optimal transport distances with regularization strength $\lambda$ between two distributions on $\mathbb R^d$ supported on $N$ points $x^{(i)},y^{(i)}$, $i=1,\dots, N$ using the Sinkhorn scaling algorithm as implemented in the Python OT package, we first construct a matrix of pairwise distances $C\in\mathbb R^{N\times N}$ where $C_{ij}=c(x^{(i)},y^{(j)})$, which requires a one-time cost of  $N^2K(d)$ operations, where $K(d)$ is the complexity of computing the cost between two $d$-dimensional vectors, then an iterative scaling procedure is performed, which consists of iterated products of the fixed matrix $e^{-C/\lambda}$ with two vector iterates so that each iteration runs in $O(N^2)$ time.  Following [1] Theorem 1, we see that the algorithm reaches its termination condition to a given precision $\eta$ (the default value in the package used is 1e-9) in a number of iterations, $K$, bounded as
> $$
> 	K\leq 2 +\frac{-4\log(e^{-\|C\|_{\infty}/\lambda }\kappa)}{\eta},
> $$
> where $\kappa$ is the smallest value in the input probability vectors (remark that this quantity is at most $\frac 1 N$ with equality when considering empirical distributions on distinct points).
>
> Given that the ratio requires two evaluations of the entropic distance, the overall complexity is twice that of the above procedure. Moreover, if the numerator is computed first we can reuse the matrix C when computing the denominator’s pairwise cost matrix.
>
> Another limitation of our method as formulated in the paper currently is that violations in each dimension are penalized equally. This may limit the utility of this methodology in applications where violations of the order in one particular dimension should be severely penalized (e.g. unsafe responses from LLMs). In such a case, one may modify the proposed framework to use a cost function of the form $c(x,y)=\sum_{i=1}^d h_i(y_i-x_i)$ (i.e. we prescribed a different cost to violations in each dimension). Provided that each of the $h_i$ are nonnegative functions satisfying the smoothness condition, all of the theoretical results derived in the paper still go through. As such, a practitioner that wishes to formulate a domain-specific notion of almost stochastic dominance can adopt a data-driven approach to learn a new stochastic order tailored to the user’s preferences by letting the $h_i$ be parametrized costs (e.g. the logistic function) and optimizing over the parameters (in the previous example, the gain).
>
> [1] Dvurechensky, P., Gasnikov, A., & Kroshnin, A. (2018, July). Computational optimal transport: Complexity by accelerated gradient descent is better than by Sinkhorn’s algorithm. In International conference on machine learning (pp. 1367-1376). PMLR.
>
> [2] Hada, R., Gumma, V., de Wynter, A., Diddee, H., Ahmed, M., Choudhury, M., ... & Sitaram, S. (2023). Are large language model-based evaluators the solution to scaling up multilingual evaluation?. arXiv preprint arXiv:2309.07462.
>
> [3] Jiang, D., Ren, X., & Lin, B. Y. (2023). Llm-blender: Ensembling large language models with pairwise ranking and generative fusion. arXiv preprint arXiv:2306.02561.
>
> [4] Liu, Y., Iter, D., Xu, Y., Wang, S., Xu, R., & Zhu, C. (2023). G-eval: Nlg evaluation using gpt-4 with better human alignment. arXiv preprint arXiv:2303.16634.
>
> [5] Zheng, L., Chiang, W.L., Sheng, Y., Zhuang, S., Wu, Z., Zhuang, Y., Lin, Z., Li, Z., Li, D., Xing, E. and Zhang, H. (2024). Judging llm-as-a-judge with mt-bench and chatbot arena. Advances in Neural Information Processing Systems, 36.

---

### Official Review · Reviewer_w1Lw · 2024-07-17

**Soundness:** 3
**Presentation:** 3
**Contribution:** 3
**Rating:** 6
**Confidence:** 2

**Summary:**

In this paper authors propose a testing framework for  First order Stochastic Dominance (FSD) for multivariate random variables. To achieve this, the authors use ideas from optimal transport using Entropic Regularization to derive a hypothesis testing procedure for testing multivariate FSD. The proposed methodology includes theoretical results showing the distributional convergence of the test statistic.

**Strengths:**

- The paper considers an important problem, which is well motivated.
- Theoretical results back the methodology
- The paper is well-presented

**Weaknesses:**

1. The experimental results do not present the Type I and II errors for the hypothesis testing methodology proposed. The authors should consider adding these results (with a synthetic setup if needed).
2. The results should also investigate how these errors change with increasing $N$? And increasing $d$?

Overall, I think more extensive empirical investigation is needed to show how fast the convergence takes place, and how well the methodology scales with increasing $d$ and $N$ (in terms of computational complexity)

**Other comments:**

Lines 166-167:
- Notation should be $OT_{h, 0}$ instead of $OT_{0, \lambda}$
- Kendall tau rank has not been defined
- From the definition of $\mathcal{E}_{\mathcal{W}_2}$  below line 73, we should have that

$\mathcal{E}_{\mathcal{W}_2}= 1$
when

$F_Y^{-1}(t) \leq F_X^{-1}(t)$ for a.e. t. However, line 77 states that in this case $\mathcal{E}_{\mathcal{W}_2} = 0$. I'm not sure if this is correct.

**Questions:**

See above

**Limitations:**

The authors discuss the limitations in the discussion section.

---

> ### Author Rebuttal · Authors · 2024-08-06
>
> We thank the reviewer for their support of this paper and for highlighting some additional experiments which would further clarify the performance of the method. We have run the proposed experiments and will add them to the appendix. We have also addressed the points from your other comments in the text.
> ___
> **The experimental results do not present the Type I and II errors for the hypothesis testing methodology proposed. The authors should consider adding these results (with a synthetic setup if needed). The results should also investigate how these errors change with increasing $N$? And increasing $d$? Overall, I think more extensive empirical investigation is needed to show how fast the convergence takes place.**
> ___
> We agree that additional experiments would help to better illustrate the strengths and limitations of this approach. In line with your recommendations, we have computed the Type I and II errors for the synthetic  experiment in the paper and conducted an empirical study of how these errors vary as a function of $N$ and $d$. The results of these experiments are compiled in the attached pdf and demonstrate that the proposed testing methodology is sample efficient and scales well even in moderate dimensions.  As mentioned, these additional experiments will be added to the appendix.
> ___
> **How well [does] the methodology scales with increasing $d$ and $N$ (in terms of computational complexity)**
> ___
>
> Thank you for bringing up this important point, we agree that the computational complexity of our approach should be more carefully discussed. We will add the following discussion to the appendix.
>
> The ratio requires computing two entropic optimal transport distances with regularization strength $\lambda$ between two distributions on $\mathbb R^d$ supported on $N$ points $x^{(i)},y^{(i)}$, $i=1,\dots,N$.
>
> To solve this problem numerically, we utilize the popular Sinkhorn scaling algorithm as implemented in the Python OT package. To compute one distance, we first construct a matrix of pairwise distances $C\in\mathbb R^{N\times N}$ where $C_{ij}=c(x^{(i)},y^{(j)})$, which requires a one-time cost of  $N^2K(d)$ operations, where $K(d)$ is the complexity of computing the cost between two $d$-dimensional vectors), then an iterative scaling procedure is performed, which consists of iterated products of the fixed matrix $e^{-C/\lambda}$ with two vector iterates so that each iteration runs in $O(N^2)$ time.  Following [1] Theorem 1, we see that the algorithm reaches its termination condition to a given precision $\eta$ (the default value in the package used is 1e-9) in a number of iterations, $K$, bounded as
> $$
> 	K\leq 2 +\frac{-4\log(e^{-||C||_{\infty}/\lambda }\kappa)}{\eta},
> $$
> where $\kappa$ is the smallest value in the input probability vectors (remark that this quantity is at most $\frac 1 N$ with equality when considering empirical distributions on distinct points).
>
> Given that the ratio requires two evaluations of the entropic distance, the overall complexity is twice that of the above procedure. Moreover, if the numerator is computed first we can reuse the matrix C when computing the denominator’s pairwise cost matrix.

---

> > ### Comment · Reviewer_w1Lw · 2024-08-13
> >
> > I would like to thank the reviewers for their response, which addresses the questions raised. I am satisfied with the response and will keep my score unchanged.

---

### Official Review · Reviewer_i6xT · 2024-07-19

**Soundness:** 3
**Presentation:** 3
**Contribution:** 3
**Rating:** 6
**Confidence:** 1

**Summary:**

The paper studies the testing of multivariate stochastic dominance, i.e., deciding an order between two multivariate random variables. The authors generalized the notion of index of almost stochastic dominance, which is for uni-variate rv. The new index is based on regularized value of optimla transport problems. Convergence of plug-in estimate of that index, and its bootstrap theory is developed. As a main application, it is applied to LLM Benchmarking, where an LLM is evaluated on many metrics.

**Strengths:**

1. This generalized notion of stochastic order index for multivariate random variables is interesting. Its application to LLM benchmarking is also interesting.

2. Many advanced tools from probability is used in the proof section. It is interesting to see that these theories find application in LLM evaluation.

**Weaknesses:**

1. There are two hyperparameters that need to be chosen to define the ratio statistics. The author should provide guidance on how to choose them.
2. Can the authors discuss the computation complexity of computing the ratio statistics?
3. The new notion of stochastic dominance is interesting. I think it would explain this new concept more if the authors could compile a list of examples that showcase this notion. For example the example in Sec 5.1 is intuitive. But it is too simplified. Since "dominance" is a complicated concept for multivariate distributions, explaining what kind of dominance is being captured and what not could be useful when practitioners apply the method.

**Questions:**

- What does $\hat \mu$ $\hat \nu$ mean in the context of LLM benchmarking? Do we divide the dataset into multiple smaller datasets and evaluate LLM on them?
- Typo at line 185. Should be "whenever $OT_{\bar h, \lambda } = 0$"

**Limitations:**

See above.

---

> ### Author Rebuttal · Authors · 2024-08-06
>
> We thank the reviewer for their careful reading of the paper and for identifying sections which could use further clarification. This feedback is invaluable to us in improving the quality of our submission. We have corrected the typos identified and will further polish the paper to improve its readability.
> ___
> **There are two hyperparameters that need to be chosen to define the ratio statistics. The author should provide guidance on how to choose them.**
> ___
> For the choice of hyperparameters, we recall that the parameter $\lambda$ corresponds to the regularization strength in the entropic optimal transport distance, which is employed as a statistically efficient and computationally tractable proxy for the standard optimal transport distance. Given Theorem 1, it is desirable to set $\lambda$ as small as possible in order to approximate the true optimal transport distance well, for example $\lambda = 0.1$ as used in the numerical experiments. We remark that for practical purposes, $\lambda$ cannot be chosen arbitrarily small due to possible underflow in the matrix $e^{-C/\lambda}$, where $C$ is the matrix of pairwise costs, used when computing the entropic distance using Sinkhorn’s algorithm (see the following point for more details). Notably, Sinkhorn’s algorithm is a method tailored to positive matrices and cannot cope with zero entries. As such, a practitioner can start with a small value for $\lambda$ and increase it gradually if numerical instability is encountered.
>
> As for the choice of function, $h$, three examples are discussed in Example 1. For practical applications, the logistic function is preferred for ease of computation and since it satisfies the desired smoothness properties. As mentioned in Example 1 and demonstrated empirically in Section 5.1, the larger the gain, $\beta$, of the logistic function is, the closer the logistic function is to being compatible with standard multivariate FSD as described in Definition 1. For the matrix $e^{-C/\lambda}$ from the previous point to be well-conditioned, there is a tradeoff between $\beta$ and $\lambda$. In the experiments, these hyperparameters were set by first fixing the entropic parameter to $\lambda = 0.1$, then $\beta$ was set as large as possible subject to a computational budget constraint. Indeed, as described in the following point of this response, the number of iterations required for Sinkhorn’s algorithm to converge scales as $||C||_{\infty}/\lambda$.
>
> We also highlight that the hyperparameter $\varepsilon_0$ used when defining the notion of entropic multivariate almost FSD does not need to be set when performing relative testing; a discussion of this approach is included on lines 252 onwards. Notably, the empirical study we performed on ranking LLMs utilizes this relative testing framework to perform the rankings using the proposed ratio statistic and does not require one to set the parameter $\varepsilon_0$.
>
> We will add these clarifications on the choice of hyperparameters to the appendix.
> ___
> **Can the authors discuss the computation complexity of computing the ratio statistics?**
> ___
> This is indeed an important question, we acknowledge that a more complete discussion is warranted. The ratio requires computing two entropic optimal transport distances with regularization strength $\lambda$ between two distributions on $\mathbb R^d$ supported on $N$ points $x^{(i)},$$y^{(i)}$ $i=1,\dots, N$.
>
> To solve this problem numerically, we utilize the popular Sinkhorn scaling algorithm as implemented in the Python OT package. To compute one distance, we first construct a matrix of pairwise distances $C\in\mathbb R^{N\times N}$ where $C_{ij}=c(x^{(i)},y^{(j)})$, which requires a one-time cost of  $N^2K(d)$ operations, where $K(d)$ is the complexity of computing the cost between two $d$-dimensional vectors), then an iterative scaling procedure is performed, which consists of iterated products of the fixed matrix $e^{-C/\lambda}$ with two vector iterates so that each iteration runs in $O(N^2)$ time.  Following [1] Theorem 1, we see that the algorithm reaches its termination condition to a given precision $\eta$ (the default value in the package used is 1e-9) in a number of iterations, $K$, bounded as
> $$
> 	K\leq 2 +\frac{-4\log(e^{-\|C\|_{\infty}/\lambda }\kappa)}{\eta},
> $$
> where $\kappa$ is the smallest value in the input probability vectors (remark that this quantity is at most $\frac 1 N$ with equality when considering empirical distributions on distinct points).
>
> Given that the ratio requires two evaluations of the entropic distance, the overall complexity is twice that of the above procedure. Moreover, if the numerator is computed first we can reuse the matrix C when computing the denominator’s pairwise cost matrix.
>
> We will add a sentence to this effect in the main text and provide a thorough discussion in the appendix.

---

> ### Author Response · Authors · 2024-08-06
> **Rebuttal continued (1/2)**
>
> ___
> **The new notion of stochastic dominance is interesting. I think it would explain this new concept more if the authors could compile a list of examples that showcase this notion. For example the example in Sec 5.1 is intuitive. But it is too simplified. Since "dominance" is a complicated concept for multivariate distributions, explaining what kind of dominance is being captured and what not could be useful when practitioners apply the method.**
> ___
> We agree that this notion of dominance should be more clearly demonstrated in terms of a more complex example. To amend this, we propose to add the following discussion to the paper.
>
> > Suppose that an agent must choose between multivariate (financial) portfolios, e.g. a list of assets from $k$ different companies, with the aim of maximizing the return. For the standard notion of stochastic dominance, it is required that each individual asset from company $i$ generally achieves a higher value than that  of company $j$; in particular, the expected return of each asset is higher. The notion of almost FSD relaxes this notion to allow for the possibility that some of the assets from company $i$ underform those of company $j$ on average, but only by a small amount.
>
> One limitation with the approach discussed in the text is that all dimensions are treated equally, as the same function $h$ is used to compare each dimension. This may limit the utility of this methodology in applications where violations of the order in one particular dimension should be severely penalized (e.g. unsafe responses from LLMs). In such a case, one may modify the proposed framework to use a cost function of the form $c(x,y)=\sum_{i=1}^d h_i(y_i-x_i)$ (i.e. we prescribed a different cost to violations in each dimension). Provided that each of the $h_i$ are nonnegative functions satisfying the smoothness condition, all of the theoretical results derived in the paper still go through. As such, a practitioner that wishes to formulate a domain-specific notion of almost stochastic dominance can adopt a data-driven approach to learn a new stochastic order tailored to the user’s preferences by letting the $h_i$ be parametrized costs (e.g. the logistic function) and optimizing over the parameters (in the previous example, the gain).
>
> We will add a sentence explaining this extension in the revised paper.
> ___
> **What does $\hat \mu,\hat \nu$ mean in the context of LLM benchmarking? Do we divide the dataset into multiple smaller datasets and evaluate LLM on them?**
> ___
>
> We acknowledge that the LLM benchmarking experiment should be more clearly explained to avoid ambiguities. The experiment is conducted as follows:
>
> 1. We use the dataset from [2], which compiles a training set of 100K prompts along with the resulting response from 12 different LLMs.
>
> 2. For each prompt, the responses from the 12 different LLMs are evaluated according to 9 automatic metrics (e.g. BLEU, ROUGE, BERTScore, BARTScore, etc) and are stored as 12 vectors in $\mathbb R^9$ $(x^{(i)})_{i=1}^{12}$, where $x^{(1)}$ is the vector containing the outputs of the 9 metrics for the 1st LLM and so on. $x^{(1)}$ is then viewed as a sample from the distribution of all possible responses from the 1st LLM evaluated according to these 9 metrics. Note that we normalize these metrics so they are all in same [0,1] range.
>
> 3. This process is repeated for each of the 100K prompts so that one can construct the empirical measure $\hat \mu_N^{(1)}=\frac{1}{N}\sum_{i=1}^N\delta_{x^{(1)}_i}$ for $N = 100K$ (here $\delta_x$ is the Dirac measure at the point $x\in\mathbb R^9$).
>
> 4. The empirical measures $\hat \mu_n^{(i)}$ and $\hat \mu_n^{(j)}$ for some $n\leq 100K$ and $i\neq j$ are then compared by computing the normalized index $\varepsilon_{h,\lambda}(\hat \mu_n^{(i)},\hat \mu_n^{(j)})=\varepsilon_{ij}^{(h,\lambda)}$.
>
> 5. Given the pairwise ratios $\varepsilon_{ij}^{(h,\lambda)}$, we rank the 12 LLMS according to the relative testing procedure described in Section 4.2. The resampling procedure (bootstrap) is used to construct the confidence intervals.
>
> 6. The results thus obtained are compared (via Kendall Tau Similarity) to a univariate FSD ranking based on ChatGPT scoring i.e. ChatGPT is presented by the instruction and the response of each LLM as described in point 1 and produces a score that judges the quality of the response in terms of following the given instruction; the different models are then ranked according to their ChatGPT score (a univariate ranking) and compared to the multivariate ranking described above.

---

> ### Author Response · Authors · 2024-08-06
> **Rebuttal continued (2/2)**
>
> We underscore that the advantage of our multivariate stochastic dominance over the ChatGPT-based scoring method or human evaluation is threefold. First, as discussed previously, it requires significantly less computational overhead. Second, there are no high upfront monetary costs associated with evaluating the ratio statistic, this can be an important consideration for large-scale comparison tasks. Third, our approach can be run locally thereby eliminating the privacy concerns of exposing sensitive data on APIs running LLM as a judge, such as ChatGPT.
> ___
>
> [1] Dvurechensky, P., Gasnikov, A., & Kroshnin, A. (2018, July). Computational optimal transport: Complexity by accelerated gradient descent is better than by Sinkhorn’s algorithm. In International conference on machine learning (pp. 1367-1376). PMLR.
>
> [2] Jiang, D., Ren, X., & Lin, B. Y. (2023). Llm-blender: Ensembling large language models with pairwise ranking and generative fusion. arXiv preprint arXiv:2306.02561.

---

### Author Rebuttal · Authors · 2024-08-06

We thank the reviewers for their careful reading of our submission. Their questions and comments have proved invaluable in improving the quality of the paper and helped us in identifying passages which were unclear and confusing. We briefly summarize the main comments brought up in the reviewers here and attach point by point detailed responses to each official review.

* Q: **The complexity of computing the violation ratio is not stated explicitly. In particular, what is its dependence on the dimension $d$ when comparing two distributions on $N$ points in $\mathbb R^d$?**

A: The entropic optimal transport distances which are used to define the ratio are computed using Sinkhorn’s algorithm. The only step of this algorithm which depends on the dimension is the construction of the pairwise cost matrix $C$ which requires a one time cost of  $N^2K(d)$ operations, where $K(d)$ is the complexity of computing the cost between two $d$-dimensional vectors. The main loop of the algorithm consists of matrix-vector products between a fixed $N\times N$ matrix and $N$-dimensional vector iterates; the number of steps required for the loop to terminate is also characterized and independent of $d$.

* Q: **How should a practitioner set the various hyperparameters, $\lambda,\beta,\varepsilon_0$?**

A: The parameter $\lambda$ serves as the regularization strength for the entropic optimal transport distance. Given that the entropic distance is used as a computationally and statistically efficient proxy for the standard optimal transport distance, it is desirable to set $\lambda$ as small as possible. As Sinkhorn’s algorithm utilizes the matrix $e^{-C/\lambda}$, the ratio $(\max_{ij} C_{ij})/\lambda$ cannot be too large, otherwise numerical underflow will be encountered. If underflow occurs, Sinkhorn’s algorithm will fail, as it can only cope with positive matrices.

As for $\beta$, the gain in the logistic cost function, it is discussed in the text and demonstrated in the experiments that $\beta$ should be taken as large as possible to best capture the notion of stochastic dominance. Given the discussion in the previous paragraph, there is a tradeoff regarding how large $\beta/\lambda$ can be for the purposes of numerical estimation. This ratio also figures in the number of Sinkhorn iterations required for convergence. As such, a practitioner may first fix a desired value of $\lambda$ (say $0.1$) and set $\beta$ to be large (but not so large as to cause underflow). If Sinkhorn iterations take too long to converge, the user may consider decreasing $\beta$ or increasing $\lambda$.

Finally, we underscore that, for the purpose of relative testing of models,  the threshold for multivariate entropic FSD, $\varepsilon_0$ does not need to be set.

* Q: **How reasonable is it to take consensus with ChatGPT as the ground truth?**

A: In the experiments, ChatGPT is being used for the purposes of evaluation as motivated by its use as a human proxy in the context of evaluating the quality of generated natural language [1-4] . It is shown that the ranking obtained via multivariate stochastic dominance on automatic metrics correlates well with the ChatGPT ranking which, in turn, correlates well with human evaluation. We underscore that our approach based on multivariate stochastic dominance is preferable to ChatGPT or human evaluation with regard to the following factors. First, as discussed previously, it requires significantly less computational overhead. Second, there are no upfront monetary costs associated with evaluating the ratio statistic, this can be an important consideration for large-scale comparison tasks. Third, our approach can be run locally thereby eliminating the privacy concerns of exposing sensitive data on APIs running LLM as a judge, such as ChatGPT.

* Q: **What notions of dominance can be captured using this methodology? If a practitioner wishes to implement a notion of stochastic ordering tailored to a particular application, can they do so using this methodology?**

One limitation of our approach is that all dimensions are treated equally as the same function $h$ is used to compare each dimension. This may limit the utility of this methodology in applications where violations of the order in one particular dimension should be severely penalized (e.g. unsafe responses from LLMs). However, the proposed framework can be adapted to use a cost function of the form $c(x,y)=\sum_{i=1}^d h_i(y_i-x_i)$ (i.e. we prescribed a different cost to violations in each dimension) provided that the $h_i$ satisfy the technical conditions described in the paper. With this, a practitioner aiming to formulate a domain-specific notion of almost stochastic dominance can adopt a data-driven approach to learn a new stochastic order tailored to the user’s preferences by letting the $h_i$ be parametrized costs (e.g. the logistic function) and optimizing over the parameters (in the previous example, the gain).


[1] Hada, R., Gumma, V., de Wynter, A., Diddee, H., Ahmed, M., Choudhury, M., ... & Sitaram, S. (2023). Are large language model-based evaluators the solution to scaling up multilingual evaluation?. arXiv preprint arXiv:2309.07462.

[2] Jiang, D., Ren, X., & Lin, B. Y. (2023). Llm-blender: Ensembling large language models with pairwise ranking and generative fusion. arXiv preprint arXiv:2306.02561.

[3] Liu, Y., Iter, D., Xu, Y., Wang, S., Xu, R., & Zhu, C. (2023). G-eval: Nlg evaluation using gpt-4 with better human alignment. arXiv preprint arXiv:2303.16634.

[4] Zheng, L., Chiang, W.L., Sheng, Y., Zhuang, S., Wu, Z., Zhuang, Y., Lin, Z., Li, Z., Li, D., Xing, E. and Zhang, H. (2024). Judging llm-as-a-judge with mt-bench and chatbot arena. Advances in Neural Information Processing Systems, 36.

---

### Author Response · Authors · 2024-08-13

Dear reviewers,

We thank you for your review and would appreciate if you can engage in acknowledging our rebuttal. We are happy to clarify any remaining questions you may have!

---

### Decision · Program_Chairs · 2024-09-25

**Decision:**

Accept (poster)

**Comment:**

There is a general consensus among reviewers that this paper, although it could benefit from a better presentation and slightly more thorough experiments, should be accepted, as it provides a novel and sound approach to the important problem of estimating the degree of stochastic dominance between multivariate distributions. I concur with this assessment and recommend acceptance.